# Improve Temporal Reasoning in Multimodal Large Language Models via Video Contrastive Decoding

**Daiqing Qi**[1]    **Dongliang Guo**[1]    **Hanzhang Yuan**[1]    **Handong Zhao**[2]

**Mengxuan Hu**[1]    **Lehan Yang**[1]    **Sheng Li**[1]

[1]University of Virginia    [2]Adobe Research

## Abstract

A major distinction between video and image understanding is that the former requires reasoning over time. Existing Video Large Language Models (VLLMs) demonstrate promising performance in general video understanding, such as brief captioning or object recognition within individual frames. However, they often struggle with temporal reasoning such as understanding continuous actions or tracking object transformations over time—which typically demands the integration of multiple frames in a temporally coherent manner. We first explore and explain such failures in Video LLMs from the perspective of *language and "image" priors*. While existing research has attempted to enhance the temporal understanding of VLLMs through various training strategies, the demand for expensive computational resources and training data often presents significant barriers. To this end, we further propose a simple yet novel idea for improving temporal reasoning in videos at no additional training cost. Specifically, to better capture the temporal structure across multiple frames—the key to effective temporal reasoning—we distort the temporal consistency in key frames *during the decoding phase*. Such corruption induces time-insensitive wrong responses from the model, which are then contrastively avoided when generating the final correct output. In this way, the model is encouraged to perform more temporally coherent reasoning. Our method yields consistent improvements across both temporal-specific and general video understanding benchmarks, demonstrating its effectiveness and generalizability.

## 1 Introduction

Benefiting from the significant advancements in Large Language Models in recent years, Video LLMs [14, 34, 35, 16, 30] have also experienced rapid development, exhibiting strong capabilities in general video understanding. A key distinction between video understanding and image understanding is that the former requires models to comprehend not only individual input frames but also the temporal relationships among them. Consequently, temporal perception is crucial for Video LLMs. However, recent studies [11, 21] have shown that even for simple and straightforward temporal reasoning questions that humans can easily answer, Video LLMs, such as LLaVA-Video [35], Video-LLaVA [14] and VILA [16, 22], often make mistakes that are clearly inconsistent with the ground truth. Recent studies [28, 8, 29, 11] have investigated the limitations of temporal reasoning in Video LLMs from a model-centric perspective. Given that a typical Video-LLM is composed of a vision encoder and an LLM backbone, a common analytical approach is to disentangle and examine the contributions of each component to temporal reasoning. While some works [28, 8, 29] attribute the temporal reasoning deficiencies of Video LLMs to ineffective video embeddings and accordingly focus on improving temporal information aggregation, others [11] leave aside the vision modules and

39th Conference on Neural Information Processing Systems (NeurIPS 2025).

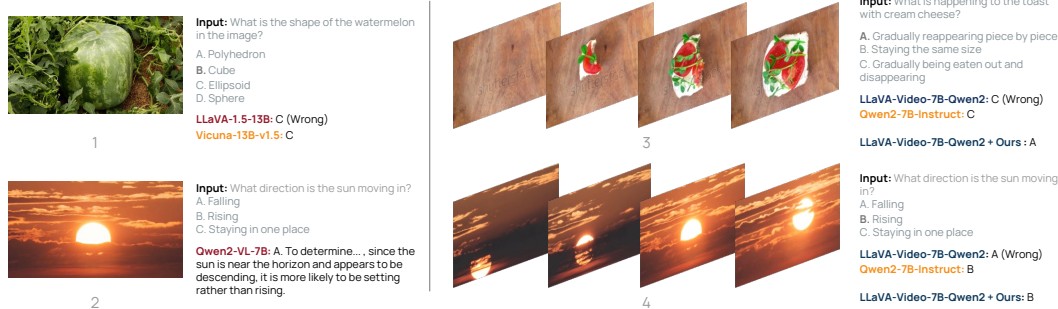

Figure 1: Example of how language and "image" priors affect Multimodal Large Language Models or Video Large Language Models in image or video understanding. (1), (3) show the negative effect of language priors and (2) reveals the how "image" priors from static visual content mislead the Video LLM's understanding of the video in (4).

embeddings, highlighting that the LLM itself exhibits weak sensitivity to the order of long textual sequences, as well as long video embeddings.

Existing works primarily focus on explaining the limitations of temporal perception from the perspective of the model itself. Consequently, the improvements derived from such analyses often require modifications to the model, such as refining video encoding [28, 8, 29] or improving the language counterpart [11]. However, these strategies typically involve modeling training (e.g., instruction tuning), which is computational- and data-intensive. Moreover, due to the increasing diversity of video LLM architectures nowadays, observations or findings from one certain architecture may not always generalize to others.

Towards this end, instead of digging into model architecture, we analyze model behavior through its responses to investigate under what circumstances video LLMs are more prone to making mistakes. We find that when video LLMs fail to understand temporal information in videos, they exhibit a behavior similar to what multimodal LLMs exhibit when hallucinate—namely, being influenced by language priors. More interestingly, due to the temporal dimension inherent in video inputs (compared to static image inputs), we observe that video LLMs also suffer from a negative influence of an *"image" prior*. For instance, the static visual features in individual frames—which do not carry temporal information, however, can mislead the LLM's perception of temporal dynamics. To avoid model-dependent modifications, we do not directly mitigating the negative effects at the model level. Instead, we take the opposite approach: we amplify these influencing factors to intentionally induce erroneous responses, and then use them as contrasting objectives in Contrastive Decoding [12]. This allows us to explicitly steer the decoding process away from such failure modes and toward generating the correct answers.

## 2 Uncovering Language and "Image" Prior in Temporal Reasoning

**What Undermines Temporal Perception in Video LLMs?** In contrast to existing works, we study the problem of limited temporal reasoning in Video-LLMs from a different perspective: instead of focusing on the model itself, we explore the conditions under which Video LLMs are more likely to struggle with temporal understanding. Although different models exhibit varying results on the same benchmark, it is observed that they tend to make mistakes more frequently on a specific subset of questions than on others. By analyzing the characteristics of these failure cases, key factors that undermine temporal reasoning can be uncovered.

*Language prior* typically refers to the prior knowledge in LLMs during (textual) pre-training or instruction-tuning, such as commonsense knowledge and reasoning in the context of multimodal understanding [17, 4]. However, it can sometimes hinder the model from grounding its answers in visual content, over-relying on the prior knowledge.

Fig. 1 (1) (top left) presents a typical example in image understanding with MLLMs. When asked about the shape of the watermelon in the image, LLaVA [19] chooses the incorrect option 'C', which aligns with the common sense that a watermelon is ellipsoidal. This strong language prior biases the

model toward selecting "C" instead of grounding to the visual input. When posing the same question without the image to its LLM backbone, Vicuna-13B-v1.5, the LLM responses with the same answer, revealing the influence of the LLM language prior on the MLLM's behavior. Similar observations are also mentioned in recent works [4] on MLLM hallucinations in image understanding. We are naturally curious whether the language prior has a similar influence on Video LLMs? If so, given the uniqueness of video inputs—such as *temporally ordered multi-frame sequences*—are there any distinct behaviors that Video LLMs may exhibit compared to MLLM with image inputs?

**Language Prior in Video LLMs.** We conducted an experiment to investigate whether language priors affect the temporal reasoning in Video LLMs. For this study, we selected two representative open-source Video LLMs—LLaVA-Video-7B-Qwen2 [35] and Video-LLaVA [13]—as test models, and used TempCompass [21] as the benchmark, which includes a variety of tasks closely related to temporal understanding, such as event ordering and attribute change. First, we performed standard inference with each Video LLM on TempCompass with both the video and textual input are provided. Then, we randomly sampled 200 questions where Video LLMs made mistakes, and conducted a blind evaluation on them, where the video input was removed and only the textual prompt was given to the LLM. The results from this blind evaluation can roughly reflect the influence of the language prior on inducing mistakes in Video LLMs.

Results show that, among the incorrect predictions made by LLaVA-Video-7B-Qwen2 and Video-LLaVA, **46.7%** and **38.9%** respectively matched the answers produced by blind LLM, significantly higher than random chance. In these cases, the Video LLMs failed to ground their responses in videos and instead relied on language priors, leading to incorrect answers.

Fig. 1 (3) provides an illustrative example. When the video is not provided, the blind LLM (Qwen2-7B-Instruct) selects option C, which corresponds to the answer with a higher prior probability that aligns with commonsense expectations. When both the video and the question are provided to the Video LLM, the model still chooses option C. This indicates that, despite access to visual input, the model remains influenced by the language prior and overlooks the temporal information present in the video. However, we observed in the experiments that in a number cases, even when the blind LLM predicted the correct answer, **the Video LLM surprisingly made an entirely different and incorrect choice** contradicting both the language prior and the information presented in the video.

**Image Prior in Video LLMs.** Fig. 1 (4) illustrates an interesting case. When the image is not provided, the blind LLM makes the right choice, which can be viewed as a positive influence of the language prior. However, the Video LLM still chooses the incorrect answer, "falling", a bias in this case introduced by the visual content of the video itself. As shown in Fig. 1 (2), when we extract a single frame from the middle of the video and input it into an MLLM (Qwen2-VL-7B) along with the question, the model perceives the scene as sunset and chooses "falling" accordingly. This visual bias ultimately leads the Video LLM to overlook the temporal progression of the sun rising and to make the wrong prediction.

Briefly, for image understanding, the input consists of both textual and visual information, and language priors can influence how MLLMs interpret visual content. In video understanding, the model must understand textual, visual, and temporal information simultaneously. In this setting, beyond language priors, visual bias—specifically, biases introduced by static frames, which we term as *"image" priors*, can also negatively impact temporal perception. As demonstrated in Fig. 1 (2), certain frames (e.g., one that resembles a sunset) may dilute the model's ability to accurately perceive temporal dynamics.

## 3 Video Temporal Distortion

Our analysis demonstrates that language and "image" priors can impair the temporal perception of Video LLMs. However, the underlying mechanisms can be more complex than they appear, potentially influenced by factors from pre-training, instruction tuning strategies, to model architecture. Instead of intervening in model design, such as the architecture or training strategies, which is often compute- and data-intensive, we seek a post-hoc correction approach. Given that priors can impair the temporal reasoning of Video LLMs, is it possible to develop a model-agnostic, plug-in method that uses the model's own temporally insensitive errors as contrastive signals, explicitly guiding the model away from such biases and increasing the probability of generating correct predictions?

*Contrastive Decoding (CD)* [12] offers a promising solution. CD is a decoding approach that aims to find text which maximizes the gap between the log-probabilities of a good and a bad LLM response. It helps us better avoid selecting tokens with high probability in the bad response, since a higher probability in the bad response means a smaller gap with its corresponding probability in the good response. Intuitively, if we can induce temporally insensitive responses from Video LLMs, they can be explicitly avoided by contrastive decoding. Therefore, the key challenge that follows is to guide Video LLMs to consistently generate such bad responses, so that they can be used as contrasting objectives during the final decoding phase.

## 3.1 Contrastive Decoding in Video Large Language Models

**Decoding in LLMs.** Considering a Video LLM parametrized by $\theta$. It takes as input a text query $\mathbf{x}$ and a video context $\mathbf{V}$, and generate a relevant response $\mathbf{y}$ to the text query. The response $\mathbf{y}$ is sampled auto-regressively from the probability distribution conditioned on the query $\mathbf{x}$ and the video context $\mathbf{V}$, formulated as:

$$\mathbf{y}_t \sim p_\theta(\mathbf{y}_t \mid \mathbf{V}, \mathbf{x}, \mathbf{y}_{<t}) \propto \exp \text{logit}_\theta(\mathbf{y}_t \mid \mathbf{V}, \mathbf{x}, \mathbf{y}_{<t}), \tag{1}$$

where $\mathbf{y}_t$ denotes the token at time step $t$, and $\mathbf{y}_{<t}$ represents generated tokens up to $(t-1)$.

**Contrastive Decoding in Video LLMs.** Specifically, given a text query $\mathbf{x}$ and a video input $\mathbf{V}$, the model generates two output distributions: one conditioned on the original $\mathbf{V}$ and the other on the distorted video input $\mathbf{V}'$, which is derived by applying pre-defined distortion (e.g., adding noise to visual features as the simplest case) to $\mathbf{V}$. Then, a new contrastive probability distribution is computed by leveraging the differences between two original distributions. The new contrastive distribution $p_{\text{vtd}}$ is formulated as:

$$p_{\text{vtd}}(\mathbf{y} \mid \mathbf{V}, \mathbf{V}', \mathbf{x}) = \text{softmax} \left[(1+\alpha)\text{logit}_\theta(\mathbf{y} \mid \mathbf{V}, \mathbf{x}) - \alpha\text{logit}_\theta(\mathbf{y} \mid \mathbf{V}', \mathbf{x})\right], \tag{2}$$

where larger $\alpha$ indicate a stronger amplification of the differences ($\alpha = 0$ reduces to regular decoding). The process is shown in Fig. 3. Wrong option "B" is assigned possibilities in the original normal distribution $\text{logit}_\theta(\mathbf{y} \mid \mathbf{V}, \mathbf{x})$ due to language or image priors. However, as temporal information is removed in the distorted video input, which further amplifies the bias, option "B" receives significantly higher scores in $\text{logit}_\theta(\mathbf{y} \mid \mathbf{V}', \mathbf{x})$. Finally, the score of "B" is notably reduced in $p_{\text{vtd}}$ after Eq. 3 is applied, leading to correct answer "A". Following Li et al. [12], we also apply an adaptive plausibility constraint on $p_{\text{vtd}}$, where CD is applied only to high-probability tokens whose probabilities exceed a fraction $\beta \in [0, 1]$ of the maximum token probability. Details are discussed in Appendix A.

## 3.2 Video Temporal Distortion

**What Makes an Effective Temporal Distortion.** Distortion strategies play a key role in elevating the probabilities of bad responses in $\text{logit}_\theta(\mathbf{y} \mid \mathbf{V}', \mathbf{x})$ while diminishing those of good responses. To consistently generate bad responses as contrasting objectives, we need *remove temporal clues in the video input while maintaining confounding priors* that induces incorrect answers neglecting temporal information. Such balance is critical. For example, in the case shown in Fig 1(4), masking the entire video unexpectedly leads to the correct answer. This is undesirable, because if the correct answer is used as a contrasting objective, its possibility is reduced with Eq. 3. What we need is an carefully distorted input, like in Fig 1(2), which lacks temporal clues and misleads the model.

Intuitive solutions include: ① adding noise to visual features ② randomly shuffling frame sequences ③ randomly dropping frames. Fig. 2 shows results of LLaVA-Video-7B [35] on EventHallusion [32], where the videos depict continuous actions that require strong temporal perception. We adopt three different strategies when applying the distortions for contrastive decoding: randomly apply

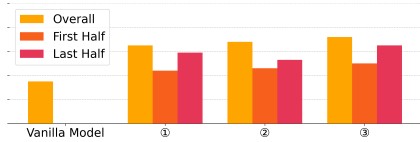

Figure 2: Distortion results.

distortion on 1. all frames, 2. first half frames and 3. last half frames. While the overall results improve, they are notably sensitive to different sampling preferences toward the beginning or the end of the video. It implies that designing adaptive distortion strategies with greater stability and adaptability is crucial. Towards this end, we propose a Video Temporal Distortion strategy that adaptively distorts frames.

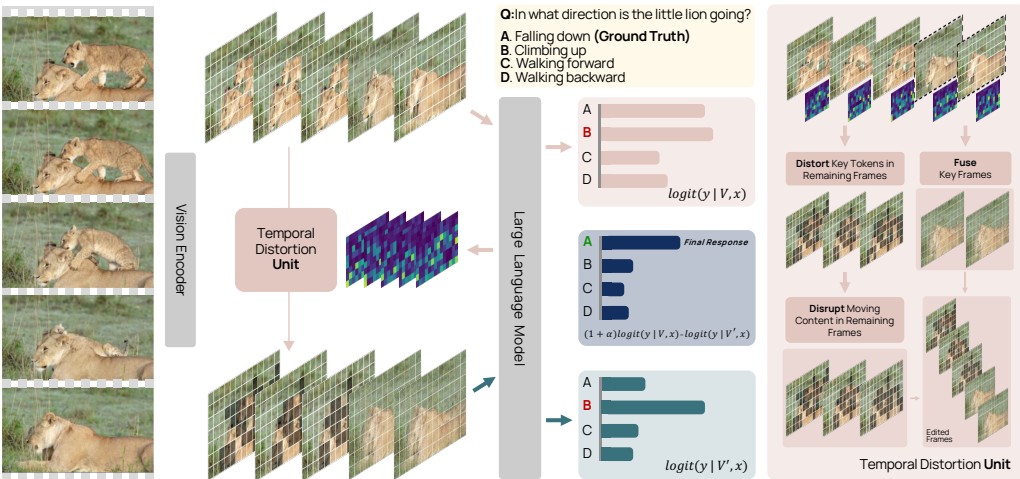

Figure 3: Overview. The original video embeddings $\mathbf{V}$ are first forwarded to the LLM with text query $\mathbf{x}$ to obtain $\mathrm{logit}_\theta(\mathbf{y} \mid \mathbf{V}, \mathbf{x})$, where intermediate attention maps in LLM layers are retrieved and fed to Temporal Distortion Unit to guide the distortion of original video embeddings. Then distorted video $\mathbf{V}'$ is input to LLM to obtain $\mathrm{logit}_\theta(\mathbf{y} \mid \mathbf{V}', \mathbf{x})$. Final token is generated from $p_{\mathrm{vtd}}$ in Eq. 3.

*Temporal Distortion Criteria.* We argue that, an ideal distortion should satisfy the following criteria: **(1)** it removes temporal information. **(2)** it includes priors that are likely to mislead the model, such as salient static visual features. **(3)** The distortion applied to the video should be adaptive towards different video inputs than random to ensure stability. Our solution is illustrated in Fig. 3. Given original video embeddings $\mathbf{V}$, and text query $\mathbf{x}$, we obtain distorted video embeddings $\mathbf{V}'$ with our Temporal Distortion Unit guided by attention maps from intermediate LLM layers to adaptively distort important frames with rich temporal information, while keeping less relevant frames that can potentially provide misleading context information or priors.

**Framework.** Fig. 3 (Right) shows the distortion pipeline. Given a set of $K$ original video frame embeddings $\{\mathbf{v}_i\}_{i=1}^K$, and corresponding attention maps $\{\mathbf{A}_i\}_{i=1}^L$ from $L$ intermediate LLM layers, We first compute the importance of each image token. To mitigate the bias of relying solely on the attention maps from the final layer, we compute *token importance* at each layer and aggregate it with momentum. Specifically, given attention map $\mathbf{A}_l$ at layer $l$, token importance matrix is computed as:

$$\mathbf{S}_l = \frac{1}{h} \sum_{i=1}^h \mathbf{A}_l^{(i,:,:)}[-1], \mathbf{A} \in \mathbb{R}^{(h,n,n)}$$

where $h$, $n$ denote the value of attention heads and unmasked input tokens so far, respectively. Consequently, the importance of an image token $\mathbf{v}_j$ indexed at $j$ is $\mathbf{S}_l[j]$ from layer $l$. To obtain the final token importance score, we apply momentum-based accumulation over all layers. Specifically, we iteratively update the aggregated importance map $\tilde{S}_l$ as follows:

$$\tilde{\mathbf{S}}_l = w_m \cdot \tilde{\mathbf{S}}_{l-1} + (1 - w_m) \cdot \mathbf{S}_l,$$

where $w_m \in [0, 1)$ is the momentum coefficient that controls the contribution of previous layers. This update emphasizes recent layer information while retaining long-range contributions from earlier layers. Then, the *frame importance* is computed as the sum of the importance scores of all image tokens within the frame.

**Key Frame Fusion.** First, we select the top-$w_{\mathrm{fdr}}$ (Frame Distortion Ratio) most important frames and remove the their temporal information by substituting each of them with the results of mean pooling over the selected set. In this way, temporal clues are removed while coarse-grained image context is retained, which can induce biased temporal insensitive inaccurate response. We further add a small amount of Gaussian noise with weight $w_{\mathrm{fpr}}$ (Frame Perturbation Ratio) to the pooled embeddings. Compared to directly dropping selected frames, mean pooling retains more confounding visual context, and is more robust than shuffling, yielding better results in practice consistently.

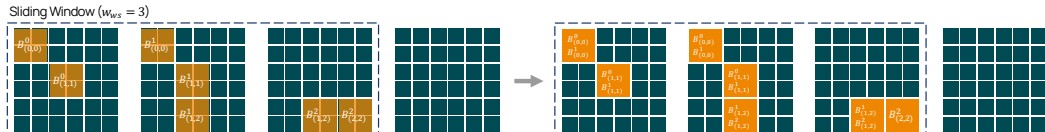

Figure 4: Disrupt moving content in remaining frames within a sliding window. **Left:** Marked dynamic blocks. **Right:** Fusion results of marked dynamics.

**Distorting Key Tokens in Remaining Frames.** Although the remaining frames are less important compared to the top-$w_{\text{fdr}}$ most important ones, their large quantity still preserves a certain degree of temporal information. If frame-level fusion is applied to these frames in the same manner as it is to key frames, it can lead to excessive information loss across the video and damage the misleading image priors, making the model unable to consistently produce the most probable mistakes. As a result, the contrasting objective becomes overly random and loses its guiding effect in contrastive decoding. Therefore, we design a more fine-grained distortion strategy specifically for this subset of frames. First, for each remaining frame, we mask the top-$w_{\text{tdr}}$ (Token Distortion Ratio) most important image tokens, thereby masking regions that potentially contain temporal cues.

**Disrupting Moving Content in Remaining Frames.** In the previous step, we performed fine-grained token-level masking on each of the remaining frames based on token importance. However, since attention maps sometimes do not accurately reflect the actual importance of each token, we instead leverage visual similarity between frames in this step to "blur" the moving content within the frames (if any), thereby removing potential temporal changes while preserving the overall image context.

Fig 4 visualizes this step. A sliding window (non-overlapping) of size $w_{\text{ws}}$ is adopted to process the entire video sequence, where we independently process the frames within a window each time. A token is considered *dynamic* if its corresponding visual content changes significantly across the window. However, due to the high spatial resolution and token granularity of popular vision encoders like CLIP [27], direct per-token comparisons are neither robust nor efficient. To address it, we first downsample each frame from size $(H, W)$ to $(\frac{H}{w_{\text{bs}}}, \frac{W}{w_{\text{bs}}})$ by applying mean pooling over non-overlapping patches, as illustrated in Fig. 4. Each resulting region is termed as *Block*.

*Downsampling.* Note that in the downsampling stage, i.e., when we downsample each frame from size $(H, W)$ to $(\frac{H}{w_{\text{bs}}}, \frac{W}{w_{\text{bs}}})$, *we do not really reduce token numbers*. As illustrated in Fig. 4 (left), we actually replace all image tokens within one "downsampled" region with the mean of the tokens included. For example, the value of each of the four token within $\mathbf{B}^0_{(0,0)}$ is the mean of the four tokens.

For a given frame $t$ and block at position $(i, j)$, we denote it as $\mathbf{B}^t_{(i,j)}$. We define its *similarity score* as the average cosine similarity with all blocks at the same position in the other frames within the window:

$$\text{sim}(\mathbf{B}^t_{(i,j)}) = \frac{1}{w_{\text{ws}}-1} \sum_{\substack{t'=1 \\ t' \neq t}}^{w_{\text{ws}}} \cos\left(\mathbf{B}^t_{(i,j)}, \mathbf{B}^{t'}_{(i,j)}\right).$$

Blocks with lower similarity scores are considered more dynamic, as they indicate greater temporal change across frames. With similarity scores for all blocks, we select the top-$w_{\text{cfr}}$ (Content Fusion Ratio) blocks with the lowest similarity scores across the entire window and label them as *Dynamic Blocks*. For each block $\mathbf{B}^t_{(i,j)}$, if it is not dynamic, we retain its original (pre-downsampled) token values. If it is dynamic and the same position across other frames in the window also contains dynamic blocks, we apply mean pooling across those positions to overwrite $B^t_{(i,j)}$ as follows.

$$\hat{\mathbf{B}}^t_{(i,j)} = \frac{1}{|\mathcal{D}_{(i,j)}|} \sum_{t' \in \mathcal{D}_{(i,j)}} \mathbf{B}^{t'}_{(i,j)},$$

where $\mathcal{D}_{(i,j)}$ is the set of frames within the window where block $(i, j)$ is marked as dynamic. Otherwise it remains unchanged. This process disrupts temporal cues by smoothing moving content while preserving static visual context details. Finally, the concatenation of the distorted key frames and the remaining frames constitutes the temporally distorted video representation $\mathbf{V}'$, which participates in contrastive decoding as defined in Eq. 3. Technical details are available in Appendix B.

| Metrics | VILA | PLLaVA | Video-ChatGPT | Video-Chat2 | LLaMA-VID | ShareGPT4-Video | LLaVA-Video | +VCD | +SID | +TCD | +Ours |
|---|---|---|---|---|---|---|---|---|---|---|---|
| Match Rate | 100.0 | 100.0 | 100.0 | 83.13 | 100.0 | 89.4 | 100.0 | 100.0 | 100.0 | 100.0 | **100.0** |
| Entire | 53.5 | 45.6 | 14.9 | 16.7 | 30.7 | 11.4 | 52.1 | 53.4 | 53.3 | 53.6 | **55.3** |
| Interleave | 62.2 | 68.9 | 79.8 | 58.6 | 98.9 | 93.7 | 60.3 | 65.7 | 65.4 | 66.8 | **75.6** |
| Misleading | 83.3 | 81.4 | 21.6 | 22.6 | 43.1 | 6.8 | 82.5 | 82.8 | 82.9 | 83.3 | **84.2** |
| Overall | 65.0 | 65.5 | 47.2 | 37.9 | 66.0 | 49.1 | 63.5 | 66.5 | 66.4 | 67.2 | **72.1** |

Table 1: EventHallusion evaluation results. Our method notably outperforms baselines on all tasks. Best results are shown in **bold**.

# 4 Experiments

**Benchmarks.** We first evaluate models on temporally-oriented benchmarks, including TempCompass [21] and EventHallusion [32]. The former targets temporal reasoning, while the latter assesses understanding of continuous actions. We then perform a comprehensive evaluation on general video understanding benchmarks, including VideoMME [5] and MLVU [36].

**Models.** We choose representative and widely-used video LLMs, Video-LLaVA [15] and LLaVA-Video-7B-Qwen2 [35], as backbone. Based on them, we apply our temporal distortion mechanism with contrastive decoding. In our experiments, we compare our method with popular Video LLMs as well as alternative contrastive decoding strategies, including VCD [9], SID [7] and TCD [32].

**Configuration.** By default, we adopt 8 frm for Video-LLaVA [15] and 32 frm for LLaVA-Video-7B-Qwen2 [35]. We run all inferences on NVIDIA A6000 GPUs and A100 GPUs. **Detailed experiment configuration and hyperparameter settings are available in Appendix C.**

## 4.1 Results

Through the experiments, we fist aim to explore the following questions: (1) Can our video temporal distortion effectively induce poor responses, thereby serving as a contrasting signal to improve temporal reasoning via contrastive decoding? (2) If so, can enhanced temporal perception also benefit general video understanding tasks? (3) Does strengthening temporal perception introduce a potential conflict with spatial understanding capabilities?

**Temporal Reasoning.** Tab. 2 shows results on TempCompass [21]. Across all four tasks and five temporally-oriented categories, our method consistently improves model performance when applied to Video-LLaVA and LLaVA-Video-7B-Qwen2. Compared to existing contrastive decoding methods such as VCD, our approach also demonstrate clear advantages. Similarly, on Tab. 1, our method yields notably better results on all tasks over baselines.

**Holistic Video Understanding.** Our method shows clear advantages on temporally-oriented benchmarks, which primarily feature videos depicting short-term, continuous changes of an event or object. In contrast, comprehensive video understanding benchmarks involve much longer videos and broader reasoning tasks across multiple aspects of video content. We evaluate LLaVA-Video-7B-Qwen2, the more powerful Video LMM, on Video-MME [5] and MLVU [36] for holistic video understanding. Results are shown on Tab. 3 and Tab. 4. Our method also demonstrates clear advantages over baselines, especially on temporally-oriented tasks, such as Temporal Perception and Reasoning in Video-MME [5] and Action Count (AC) and Action Order (AO) in MLVU [36].

## 4.2 Analysis

**Ablations.** We also explore the contribution of each component in our method, as well as how hyperparameters affect model performance. Tab. 5 demonstrates the effectiveness of each module in Temporal Distortion Unit, and the importance of sampling key frames with attention guidance. In Fig. 5, we show how each hyperparameter in Contrastive Decoding (CD) and Temporal Distortion Unit (TDU) influences model performance on TempCompass with LLaVA-Video-7B-Qwen2 [35].

From Fig. 5, we observe that a moderate level of distortion is crucial for effective contrastive decoding. In the ablation study of parameters most closely related to the degree of distortion—such as Frame Distortion Ratio, Token Distortion Ratio, Content Fusion Ratio, and again Frame Distortion

| | Model | Intern-VL2 [2] | LLaVA [10] OneVision | Long VA [33] | VILA [16] | VID-LLaVA | +VCD | +SID | +TCD | +Ours | LLaVA-Video | +VCD | +SID | +TCD | +Ours |
|---|---|---|---|---|---|---|---|---|---|---|---|---|---|---|---|
| **Multi-Choice QA** | Action | 93.8 | 96.5 | 92.3 | 92.9 | 76.0 | 76.9 | 77.0 | 77.2 | **78.4** +2.4 | 95.6 | 96.4 | 96.4 | 96.3 | **96.7** +1.1 |
| | Direction | 43.9 | 40.6 | 36.7 | 33.7 | 35.2 | 35.8 | 35.8 | 35.9 | **36.7** +1.5 | 40.3 | 41.3 | 41.6 | 41.6 | **43.6** +3.3 |
| | Speed | 51.1 | 45.4 | 43.2 | 44.1 | 35.6 | 37.0 | 37.1 | 37.4 | **38.6** +3.0 | 50.5 | 49.2 | 49.6 | 49.9 | **51.7** +1.2 |
| | Event Order | 67.2 | 69.5 | 54.3 | 50.0 | 37.7 | 39.0 | 39.1 | 39.4 | **40.4** +2.7 | 71.2 | 71.8 | 71.2 | 69.9 | **72.5** +1.3 |
| | Attr. Change | 59.9 | 56.9 | 52.4 | 60.0 | 40.9 | 42.0 | 42.2 | 42.1 | **43.8** +2.9 | 71.5 | 72.5 | 72.7 | 72.8 | **75.3** +3.8 |
| | Average | 65.5 | 64.8 | 56.1 | 56.4 | 45.5 | 46.5 | 46.6 | 46.8 | **48.1** +2.6 | 65.8 | 66.1 | 66.2 | 66.1 | **68.0** +2.2 |
| **Yes/No QA** | Action | 84.8 | 86.0 | 86.2 | 84.8 | 74.3 | 75.2 | 75.4 | 75.4 | **76.1** +1.8 | 87.7 | 88.4 | 88.7 | 88.9 | **90.4** +2.7 |
| | Direction | 53.2 | 55.3 | 50.4 | 52.2 | 51.8 | 52.5 | 52.4 | 52.7 | **53.8** +2.0 | 54.3 | 54.9 | 54.6 | 55.1 | **56.7** +2.4 |
| | Speed | 61.3 | 57.4 | 53.1 | 54.3 | 50.2 | 51.2 | 51.2 | 51.4 | **53.3** +3.1 | 58.1 | 58.9 | 59.8 | 59.4 | **62.1** +4.0 |
| | Event Order | 70.7 | 76.2 | 61.8 | 61.2 | 49.2 | 49.2 | 49.3 | 49.2 | **49.7** +0.5 | 67.2 | 66.4 | 66.5 | 66.5 | **67.4** +0.2 |
| | Attr. Change | 63.0 | 59.1 | 54.5 | 61.3 | 51.1 | 52.0 | 51.9 | 52.1 | **53.2** +2.1 | 63.6 | 63.7 | 63.8 | 63.7 | **65.8** +2.2 |
| | Average | 68.2 | 69.7 | 62.1 | 63.6 | 56.3 | 57.1 | 57.1 | 57.2 | **58.3** +2.0 | 66.8 | 67.4 | 67.5 | 67.4 | **69.4** +2.6 |
| **Matching** | Action | 96.6 | 96.0 | 94.6 | 95.0 | 87.9 | 88.0 | 88.0 | 88.2 | **89.1** +1.2 | 96.0 | 96.2 | 96.4 | 96.0 | **96.6** +0.6 |
| | Direction | 59.9 | 56.9 | 54.4 | 58.7 | 53.8 | 53.8 | 53.9 | 54.1 | **55.2** +1.4 | 59.3 | 59.6 | 59.7 | 59.9 | **60.9** +1.6 |
| | Speed | 67.0 | 61.9 | 53.3 | 60.5 | 58.4 | 58.6 | 58.9 | 58.8 | **59.9** +1.5 | 61.6 | 61.8 | 61.9 | 61.9 | **62.3** +0.7 |
| | Event Order | 84.0 | 81.3 | 64.3 | 66.0 | 59.0 | 61.3 | 61.9 | 62.2 | **65.0** +6.0 | 71.7 | 75.4 | 75.8 | 76.0 | **80.7** +9.0 |
| | Attr. Change | 77.1 | 73.8 | 62.5 | 65.3 | 58.3 | 59.8 | 59.7 | 59.4 | **63.5** +5.2 | 71.5 | 73.9 | 74.0 | 74.2 | **77.1** +5.6 |
| | Average | 77.1 | 73.8 | 65.7 | 68.9 | 63.3 | 64.1 | 64.3 | 64.3 | **66.4** +3.1 | 71.8 | 73.1 | 73.2 | 73.2 | **75.2** +3.4 |
| **Generation** | Action | 84.6 | 79.3 | 75.8 | 74.7 | 50.8 | 51.4 | 51.5 | 51.7 | **53.3** +2.5 | 85.5 | 86.1 | 85.9 | 85.7 | **87.9** +2.4 |
| | Direction | 38.8 | 30.7 | 35.3 | 36.2 | 28.7 | 29.2 | 29.0 | 29.3 | **30.3** +1.6 | 39.8 | 40.4 | 40.6 | 40.7 | **41.6** +1.8 |
| | Speed | 31.2 | 25.3 | 32.2 | 31.7 | 23.2 | 24.0 | 24.3 | 24.1 | **25.5** +2.3 | 32.0 | 33.1 | 32.7 | 33.0 | **34.1** +2.1 |
| | Event Order | 60.8 | 56.8 | 35.3 | 46.7 | 38.2 | 38.3 | 38.5 | 38.6 | **39.9** +1.7 | 61.7 | 62.1 | 61.9 | 62.2 | **63.1** +1.4 |
| | Attr. Change | 60.2 | 57.4 | 45.8 | 47.1 | 33.6 | 33.9 | 34.1 | 33.9 | **35.6** +2.0 | 61.3 | 62.2 | 62.3 | 61.9 | **63.5** +2.2 |
| | Average | 52.1 | 47.6 | 44.7 | 47.1 | 34.8 | 35.3 | 35.4 | 35.4 | **36.9** +2.1 | 55.8 | 56.5 | 56.4 | 56.5 | **57.8** +2.0 |
| **Avg. Category** | Action | 84.8 | 86.0 | 86.4 | 85.9 | 71.4 | 72.8 | 72.9 | 73.1 | **74.2** +2.8 | 91.2 | 91.7 | 91.8 | 91.7 | **93.0** +1.8 |
| | Direction | 53.2 | 55.3 | 44.2 | 45.3 | 42.4 | 42.8 | 42.7 | 43.0 | **44.0** +1.6 | 48.4 | 49.1 | 49.1 | 49.2 | **50.7** +2.3 |
| | Speed | 61.3 | 57.4 | 45.8 | 47.7 | 41.9 | 42.7 | 42.8 | 42.9 | **44.3** +2.4 | 50.5 | 50.7 | 50.9 | 50.9 | **52.6** +2.1 |
| | Event Order | 70.7 | 76.2 | 53.0 | 55.6 | 45.7 | 46.9 | 47.2 | 47.3 | **48.8** +3.1 | 68.0 | 68.9 | 68.8 | 68.7 | **70.9** +2.9 |
| | Attr. Change | 63.0 | 59.1 | 53.3 | 58.0 | 45.7 | 46.9 | 46.9 | 46.8 | **49.0** +3.3 | 67.0 | 68.1 | 68.2 | 68.1 | **70.4** +3.4 |
| | Overall | 66.0 | 64.2 | 56.9 | 58.8 | 49.8 | 50.4 | 50.5 | 50.6 | **52.1** +2.3 | 65.0 | 65.7 | 65.8 | 65.7 | **67.5** +2.5 |

Table 2: Video temporal understanding evaluation on TempCompass. Best results are shown in **bold**.

| Method | Temporal Perception | Temporal Reasoning | Short | Medium | Long | Overall |
|---|---|---|---|---|---|---|
| Video-LLaVA-7B [15] | - | - | 45.3 | 38.0 | 36.2 | 39.9 |
| LLaVA-NeXT-Video-7B-DPO [18] | 40.0 | 29.4 | 48.9 | 42.0 | 35.6 | 42.1 |
| Llama-3-VILA1.5-8B [16] | 50.9 | 41.2 | 56.1 | 42.1 | 39.6 | 45.9 |
| VILA1.5-40B [16] | 60.0 | 40.7 | 72.0 | 61.2 | 53.8 | 62.3 |
| InternVL-Chat-V1.5-20B [3] | 45.5 | 33.3 | 60.2 | 46.4 | 45.6 | 50.7 |
| LongVA-7B [33] | 58.2 | 37.3 | 61.1 | 50.4 | 46.2 | 52.6 |
| LLaVA-Video-7B-Qwen2 [35] | 61.1 | 54.3 | 73.5 | 54.8 | 51.1 | 59.8 |
| + VCD [9] | 75.4 | 55.3 | 73.9 | 55.2 | 50.8 | 60.0 |
| + SID [7] | 75.3 | 55.4 | 74.0 | 55.1 | 50.6 | 59.9 |
| + TCD [32] | 75.9 | 55.8 | 73.6 | 55.1 | 50.9 | 59.8 |
| **+ Ours** | **84.1** | **57.8** | **75.3** | **57.2** | **52.2** | **61.6** |

Table 3: Video-MME evaluation results. Our method enhances LLaVA-Video accuracy across various video durations, even outperforming VILA1.5-40B in temporal reasoning. Best results are shown in **bold**.

Ratio—we find that setting the values too low results in limited improvements, while excessively high values, i.e., severe distortion, lead to a relative decline in performance. This aligns with our earlier analysis: overly severe distortion tends to randomize the model's responses, thereby undermining its role as a negative response to guide the generation in contrastive decoding. Only appropriately calibrated distortion can effectively induce negative responses, thereby enhancing performance via contrastive decoding.

**Is There a Trade-off Between Temporal and Spatial Understanding?** Intuitively, when humans watch a video and focus on temporal changes in the scene, their attention to static details tends to decline. Interestingly, we observe a similar phenomenon in Video LLMs. On the MLVU benchmark, while our performance improves notably on temporally related tasks such as AC and AO, it drops on

| Model | AC* | ER | Needle QA | AO* | Plot QA | AR | TR | Overall |
|---|---|---|---|---|---|---|---|---|
| Video-ChatGPT-7B [23] | 31.1 | 42.0 | 40.3 | 25.1 | 29.9 | 24.0 | 26.9 | 31.3 |
| Video-LLaVA-7B [31] | 35.9 | 45.2 | 53.2 | 20.1 | 48.4 | 57.0 | 71.6 | 47.3 |
| MA-LMM-7B [6] | 24.3 | 38.9 | 43.1 | 25.1 | 35.8 | 35.5 | 51.9 | 36.4 |
| Llama-3-VILA1.5-8B [16] | 0.0 | 24.7 | 32.4 | 6.6 | 20.0 | 27.0 | 46.2 | 22.4 |
| VILA1.5-40B [16] | 11.7 | 35.8 | 38.3 | 34.3 | 62.0 | 56.4 | 84.7 | 46.2 |
| InternVL-Chat-V1.5-20B [3] | 13.3 | 24.5 | 40.0 | 14.3 | 42.0 | 51.3 | 80.2 | 37.9 |
| LongVA-7B [33] | 25.2 | 48.6 | 70.4 | 41.7 | 68.1 | 58.5 | 82.2 | 56.4 |
| LLaVA-Video-7B-Qwen2 [35] | 41.8 | 68.5 | 76.3 | 57.9 | 75.1 | **67.8** | **84.5** | 67.4 |
| + VCD [9] | 42.8 | 68.5 | 77.0 | 60.1 | 75.1 | 65.1 | 82.8 | 67.3 |
| + SID [7] | 42.9 | 68.5 | 77.2 | 60.1 | 75.1 | 65.2 | 82.7 | 67.4 |
| + TCD [32] | 42.3 | 68.6 | 76.7 | 60.2 | 75.2 | 65.0 | 83.0 | 67.3 |
| **+ Ours** | **44.1** | **68.8** | **78.5** | **62.6** | **75.8** | 65.7 | 83.8 | **68.5** |

Table 4: MLVU evaluation results. Our method achieves the best overall performance, with notable gains in temporal-related aspects. TR: Topic Reasoning, AR: Anomaly Recognition, ER: Ego Reasoning, AO: Action Order, AC: Action Count. * denotes temporal-related dimensions. Best results are in **bold**.

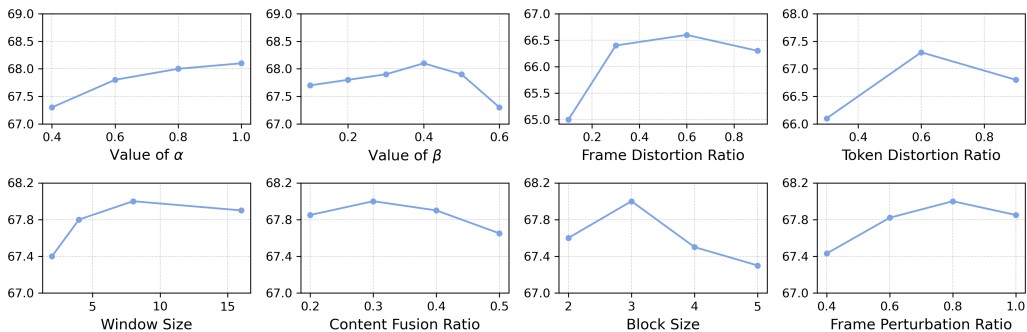

Figure 5: Sensitivity to Hyperparameter Settings on TempCompass.

AR and TR. We further conduct an experiment on TempCompass, where we progressively increase the Token Distortion Ratio and the Frame Distortion Ratio, respectively. As shown in the line plot, the accuracy of the sub-task "Attribute Change" keeps increasing, indicating an improvement in temporal perception. However, the overall accuracy (represented by the bar chart) begins to decline when the distortion ratio is approximately 0.6, suggesting that other sub-tasks are negatively affected.

| Method | TempCompass | EventHallusion |
|---|---|---|
| Vanilla | 67.5 | 72.1 |
| - Attention Guidance | 65.7 | 67.1 |
| - Key Frame Fusion | 66.1 | 66.3 |
| - Key Token Distortion | 66.5 | 68.3 |
| - Moving Content Disruption | 66.9 | 69.7 |

Table 5: Ablation Study Results on Temporal Benchmarks: TempCompass and EventHallusion.

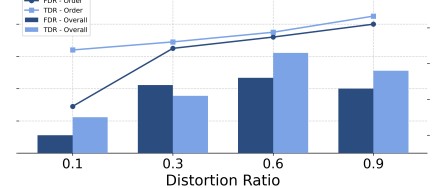

Figure 6: Influence of Distortion

## 5   Related Work

**Video Large Language Models.** Multimodal Large Language Models (MLLM) [20, 25, 26] are evolving rapidly, advancing image-text dialogue through fine-tuning pre-trained Large Language Models (LLM) with image features from additional visual encoders. Video LLMs extend MLLMs from image to video understanding by involving encoded video frames during training, such as Video-LLaVA [14], VILA-series [22, 1, 16] and LLaVA-NeXT-video [34]. To include stronger temporal information in video representations, recent works [31, 24] design additional temporal-aware encoders or customize their own training data with time-stamp annotations. Similarly, Li et al. [11] also improve

temporal reasoning with additional data but sourced from text. In contrast to existing works, our method enhances temporal reasoning in videos by incorporating a new temporal-aware decoding strategy to avoid time-insensitive responses at no additional training cost.

**Temporal Understanding in Video LLMs.** Temporal understanding is fundamental in Video LLMs. While existing popular benchmarks such as Video-MME [5] and MLVU [36] focus on the general evaluation of Video LLMs across diverse video categories and durations, TempCompass [21] specifically focuses on the temporal reasoning ability of Video LLMs with a variety of temporally focused tasks. EventHallusion [32] investigates Video LLMs' capability to understand continuous events. Our method not only achieves notable improvements on temporal understanding tasks, but also demonstrates promising results on general video benchmarks.

**Contrastive Decoding.** Contrastive Decoding (CD) [12] is a search-based LLM decoding approach that improves text generation by explicitly avoiding poor responses during decoding. Recent works [9, 7] explore its application in reducing hallucination in MLLMs by deliberately introducing distorted image inputs to elicit poor responses, which are then used to guide the model away from such errors. TCD [32] inherits similar idea to avoid video event hallucination by randomly dropping frames. Different from existing works, our work particularly focuses on improving temporal reasoning in Video LLMs. Extended discussions are available in Appendix E.

## 6   Limitations

When performing video distortion, our Temporal Distortion Unit relies solely on signals from the model itself—specifically, the attention maps extracted from the intermediate LLM layers—as guidance to estimate the importance of each visual token and each video frame. Compared to treating all frames equally and applying uniform random sampling, our approach represents a significant improvement. However, it is still not perfect. Attention maps do not always accurately reflect the true importance of each visual token, and relying on them often yields only coarse-grained results. To more precisely assess the importance of visual representations, future work may explore more accurate and robust methods beyond attention-based guidance.

Moreover, our current study is limited to Video LLMs, with distortion applied only to the visual representations. In practice, many videos come with accompanying subtitles, and models often take both video and subtitle inputs. An interesting future direction would be to distort both modalities—applying not only visual distortion but also video-aware distortion to subtitles. This would be challenging and different from the purely text-based distortion strategies employed in existing works on contrastive decoding for LLMs.

## 7   Conclusion

In this work, we investigated the challenges of temporal reasoning in Video Large Language Models (Video LLMs) and identified two key factors contributing to their failures—language prior and image prior. Building on these insights, we proposed video contrastive decoding with temporal distortion, a simple yet effective method that enhances temporal coherence without requiring additional training or computational overhead. By intentionally introducing temporal distortions in key frames and contrastively optimizing against such failures, our method encourages models to maintain temporal consistency and avoid time-insensitive predictions. Extensive experiments demonstrate that our approach significantly improves both temporal-specific and general video understanding benchmarks, showing strong effectiveness, generalizability, and scalability for improving temporal reasoning in multimodal large language models.

## Acknowledgment

The work is supported in part by the U.S. Office of Naval Research Award under Grant Number N00014-24-1-2668, the National Science Foundation under Grants IIS-2316306 and CNS-2330215, the National Institutes of Health (NIH) under Grant R01EB293388, and gifts from Adobe Research.

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

# A  Adaptive Plausibility Constraint

**Contrastive Decoding in Video Large Language Models.** Given a text query $\mathbf{x}$ and a video input $\mathbf{V}$, the model generates two output distributions: one conditioned on the original $\mathbf{V}$ and the other on the distorted video input $\mathbf{V}'$, which is derived by applying pre-defined distortion (e.g., adding noise to visual features as the simplest case) to $\mathbf{V}$. Then, a new contrastive probability distribution is computed by leveraging the differences between two original distributions. The new contrastive distribution $p_{\text{vtd}}$ is formulated as:

$$p_{\text{vtd}}(\mathbf{y} \mid \mathbf{V}, \mathbf{V}', \mathbf{x}) = \text{softmax}\left[(1 + \alpha)\text{logit}_\theta(\mathbf{y} \mid \mathbf{V}, \mathbf{x}) - \alpha\text{logit}_\theta(\mathbf{y} \mid \mathbf{V}', \mathbf{x})\right], \tag{3}$$

where larger $\alpha$ indicate a stronger amplification of the differences ($\alpha = 0$ reduces to regular decoding).

**Adaptive Plausibility Constraint.** Eq. 3 rewards texts favored by the response with original video inputs and penalizes texts favored by the response with distorted video inputs. However, the response with distorted video inputs is not always mistaken. Although video inputs are distorted, they may still preserve useful information, which can lead to correct answers. Therefore, penalizing all texts from response with distorted video inputs indiscriminately would penalize these correct answers, and conversely reward implausible answers. To tackle this issue, we follow Li et al. [12] to introduce the plausibility constraint.

Adaptive plausibility constraint is contingent upon the confidence level associated with the output distribution with original video inputs:

$$\mathcal{V}_{\text{head}}(\mathbf{y}_{<t}) = \left\{\mathbf{y}_t \in \mathcal{V} : p_\theta(\mathbf{y}_t \mid \mathbf{V}, \mathbf{x}, \mathbf{y}_{<t}) \geq \beta \max_{\mathbf{w}} p_\theta(\mathbf{w} \mid \mathbf{V}, \mathbf{x}, \mathbf{y}_{<t})\right\} \tag{4}$$

$$p_{\text{vtd}}(\mathbf{y}_t \mid \mathbf{V}, \mathbf{V}', \mathbf{x}) = 0, \quad \text{if } \mathbf{y}_t \notin \mathcal{V}_{\text{head}}(\mathbf{y}_{<t}) \tag{5}$$

where $\mathcal{V}$ is the output vocabulary of LVLMs and $\beta$ is a hyperparameter in $[0, 1]$ for controlling the truncation of the next token distribution. Larger $\beta$ indicates more aggressive truncation, keeping only high-probability tokens.

Combining the video contrastive decoding and the adaptive plausibility constraint, we obtain the full formulation:

$$\mathbf{y}_t \sim \text{softmax}\left[(1 + \alpha)\text{logit}_\theta(\mathbf{y}_t \mid \mathbf{V}, \mathbf{x}, \mathbf{y}_{<t}) - \alpha\text{logit}_\theta(\mathbf{y}_t \mid \mathbf{V}', \mathbf{x}, \mathbf{y}_{<t})\right] \tag{6}$$
$$\text{subject to } \mathbf{y}_t \in \mathcal{V}_{\text{head}}(\mathbf{y}_{<t})$$

# B  Technical Details of Video Temporal Distortion

## B.1  Disrupting Moving Content in Remaining Frames

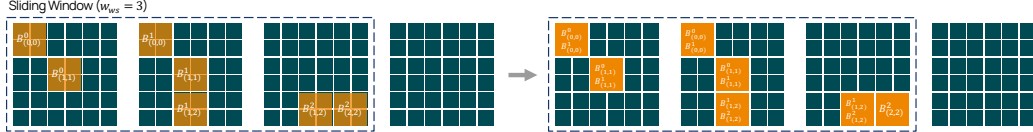

Figure 7: Disrupt moving content in remaining frames within a sliding window. **Left:** Marked dynamic blocks. **Right:** Fusion results of marked dynamics.

**Downsampling.** Note that in the downsampling stage, i.e., when we downsample each frame from size $(H, W)$ to $(\frac{H}{w_{\text{bs}}}, \frac{W}{w_{\text{bs}}})$, *we do not really reduce token numbers*. As illustrated in Fig. 7 (left), we actually replace all image tokens within one "downsampled" region with the mean of the tokens included. For example, the value of each of the four token within $\mathbf{B}^0_{(0,0)}$ is the mean of the four tokens.

## C Experiments

### C.1 Experimental Configuration

In **TempCompass** [21], we use the following hyperparameters: $\alpha = 1$, $\beta = 0.2$, $w_{\text{fdr}} = 0.2$, $w_{\text{tdr}} = 0.4$, $w_{\text{ws}} = 8$, $w_{\text{cfr}} = 0.3$, $w_{\text{bs}} = 3$, $w_{\text{fpr}} = 0.5$, $w_{\text{momentum}} = 0.8$. In **EventHallusion** [32], we use the following hyperparameters: $\alpha = 1$, $\beta = 0.2$, $w_{\text{fdr}} = 0.5$, $w_{\text{tdr}} = 0.5$, $w_{\text{ws}} = 8$, $w_{\text{cfr}} = 0.3$, $w_{\text{bs}} = 3$, $w_{\text{fpr}} = 0.8$, $w_{\text{momentum}} = 0.8$. In **Video-MME** [5] and **MLVU** [36], we use the following hyperparameters: $\alpha = 1$, $\beta = 0.2$, $w_{\text{fdr}} = 0.6$, $w_{\text{tdr}} = 0.8$, $w_{\text{ws}} = 8$, $w_{\text{cfr}} = 0.3$, $w_{\text{bs}} = 3$, $w_{\text{fpr}} = 0.8$, $w_{\text{momentum}} = 0.5$.

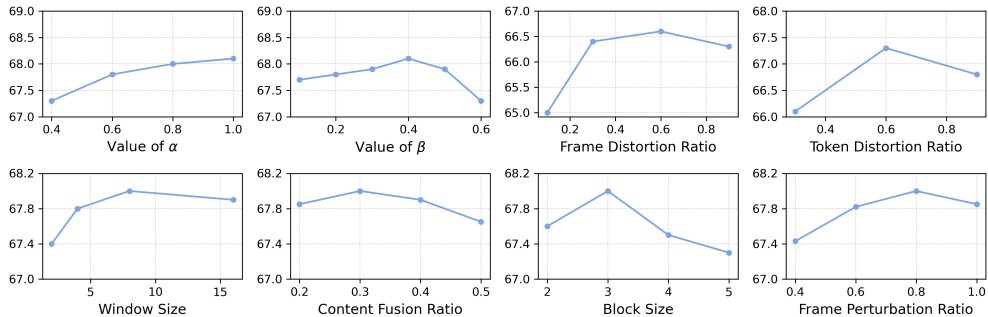

Figure 8: Sensitivity to Hyperparameter Settings on TempCompass [21].

### C.2 Analysis

From Fig. 8, we observe that a moderate level of distortion is crucial for effective contrastive decoding. In the ablation study of parameters most closely related to the degree of distortion—such as Frame Distortion Ratio, Token Distortion Ratio, Content Fusion Ratio, and again Frame Distortion Ratio—we find that setting the values too low results in limited improvements, while excessively high values, i.e., severe distortion, lead to a relative decline in performance. This aligns with our earlier analysis: overly severe distortion tends to randomize the model's responses, thereby undermining its role as a negative response to guide the generation in contrastive decoding. Only appropriately calibrated distortion can effectively induce negative responses, thereby enhancing performance via contrastive decoding.

## D Extended Discussion

We are among the first to explore video temporal understanding from the perspective of language and image priors, and to enhance it using contrastive decoding with video temporal distortion.

We are among the first to explore video temporal understanding from the perspective of language and image priors, and to enhance it using contrastive decoding with video temporal distortion.

Recent works [9, 7] have applied contrastive decoding to mitigate hallucinations in image understanding with MLLMs. For example, VCD [9] introduces random noise to distort the original image, while SID [7] prunes important tokens based on attention guidance. TCD [32] alleviates event hallucination in videos by randomly dropping frames.

SID [7] adopts a similar strategy to estimate token importance and removes the most important tokens—this is conceptually similar to the second step of our video temporal distortion. However, there are notable differences in how attention maps are utilized and how the pruning is applied. Specifically, SID uses attention maps from the $k$-th layer to assess token importance and then prunes the most important tokens starting from the $(k + 1)$-th layer.

In contrast, our approach aggregates attention maps from all layers, from shallow to deep, to compute a more accurate importance score. Furthermore, while SID performs pruning from intermediate layers, we input the distorted video representations directly at the first layer, ensuring that the dropped

information is effectively masked from the very beginning. This design allows our method to better mask information that should be dropped.

# E Efficiency

Due to the nature of Contrastive Decoding (which requires two forward passes), CD-based methods (e.g., VCD and ours) are inevitably slower than the vanilla model in generation speed.

In practical scenarios, it's difficult to optimize performance, time efficiency, and memory efficiency all at once. Different applications prioritize different aspects. CD-based methods are less time-efficient, but they offer better performance without a significant increase in memory usage. This makes them well-suited for applications where content quality is critical, such as education and medical assistance, where accuracy really matters and errors come at a high cost.

**Time-Efficient Implementation**. In practice, we can optimize the code to reduce the time by nearly half with no impact on the results, making it as efficient as the vanilla Video LLM.

The implementation is simple: we parallelize the two forward passes during next-token generation. Previously, it is implemented in the `def sample()` function as follows:

```
outputs = self.forward(...)        # inference with raw video
outputs_cd = self.forward(...)     # inference with distorted video

# calculate final logits with outputs and outputs_cd:
...
```

Now, with just a few extra lines, we use `torch.cuda.Stream()` to run both forward passes in parallel:

```
stream_main = torch.cuda.Stream()
stream_cd = torch.cuda.Stream()
outputs_holder = {}
outputs_cd_holder = {}

# submit main inference
with torch.cuda.stream(stream_main):
    outputs_holder['main'] = self.forward(...)

# submit contrastive inference
with torch.cuda.stream(stream_cd):
    outputs_cd_holder['cd'] = self.forward(...)

torch.cuda.synchronize()

outputs = outputs_holder['main']
outputs_cd = outputs_cd_holder['cd']

# calculate final logits with outputs and outputs_cd:
...
```

With this implementation, our method can be as fast as the vanilla Video LLM with the same average GPU memory usage. Note that the peak GPU memory usage will be larger.

