# OpenReview forum: "Improve Temporal Reasoning in Multimodal Large Language Models via Video Contrastive Decoding"
_NeurIPS.cc/2025/Conference — NeurIPS 2025 poster_

### Official Review · Reviewer_SAQe · 2025-06-20

**Clarity:** 3
**Significance:** 2
**Originality:** 2
**Rating:** 4
**Confidence:** 3

**Summary:**

This paper proposes a plug-in, model-agnostic method to enhance the temporal reasoning capabilities of Video Large Language Models at inference time. Building on Contrastive Decoding, the authors introduce a Temporal Distortion Unit that adaptively corrupts key frames or tokens, guided by intermediate LLM attention map, to generate a “bad” video embedding. By contrasting the original logits with those from the distorted input, the decoding process down-weights temporally insensitive errors and boosts correct predictions
. The approach is evaluated on temporally focused benchmarks (TempCompass, EventHallusion) and broad video benchmarks (Video-MME, MLVU), demonstrating consistent gains over baseline Video-LLaVA and LLaVA-Video-7B-Qwen2 models .

**Questions:**

1. See weakness section.

2. Can you provide inference latency and GPU usage metrics to quantify the runtime overhead of TDU + CD compared to vanilla decoding?

3.  Fig. 5 shows performance fluctuates as distortion ratios (wfdr, wtdr, wcfr, etc.) and α vary. Does the proposed method requires need significant per-task tuning?

**Ethical Concerns:**

["NO or VERY MINOR ethics concerns only"]

**Final Justification:**

As written in my response to the authors rebuttal.  I find this paper has significant limitations and little potential impact. I would like to keep my rating.

**Limitations:**

Yes

**Quality:**

2

**Strengths And Weaknesses:**

**Strengths**

1. Model-agnostic post-hoc correction requires no additional training, making it easy to apply to any Video LLM.

2. Outperforms prior contrastive methods (VCD, SID, TCD) by 1–3 pts on temporal tasks (e.g., Action Count, Action Order in MLVU)

3. Component-level analysis (attention guidance, key-frame fusion, token distortion, content disruption) shows each module’s contribution, and Fig. 5 illustrates sensitivity to hyperparameters

**Weakness**

1. The method largely follows the existing visual CD framework; the only substantive innovation is the specific design of the TDU. Beyond repurposing CD for video, there is limited new insight.

2. Experiments are restricted to two 7 B Video LLMs (Video-LLaVA and LLaVA-Video-7B-Qwen2). Those models are relatively old. No evaluation on other sota Video LLMs such as internvl3 or qwenvl 2.5

3. I am not sure if this method will scale to larger models such as 30B. There should not be high barrier from trying larger models (30B) nowadays given the inference-only nature of the proposed method and the advancement of highly optimized inference libraries (sglang & vllm)

4. Contrastive decoding with TDU requires at least two forward passes per token and additional distortion computation. The paper reports accuracy gains but omits any runtime, memory, or GPU-hour comparisons; a 2–3 % boost may not justify the extra cost in real-time settings. For example, a larger models (14B) often yield ≥ 2 % improvements out-of-the-box compared to a 7B model.

---

> ### Author Rebuttal · Authors · 2025-07-30
>
> Thank you for your review and suggestions!
>
> ## Weakness 1
>
> Our substantive innovation is not only the TDU, but also our unique analysis of temporal understanding with CD. Specifically:
>
> 1. An investigation into the distinct factors, language and image priors, that affect temporal reasoning in videos **(Sec. 2)**; and
> 2. Three critical conditions when applying CD to effectively counteract biases introduced by these priors **(Sec. 3.2, L155–L182)**.
>
> Substantially, most recent existing (visual or textual) CD approaches share the same principle: *distort inputs in a way to induce bad outputs as negative targets*. The novelty and interesting part is not the principle itself but lies in analyzing what factors are critical for reasoning in different tasks **(Sec. 2)**, and how to manipulate them effectively **(Sec. 3.2)**.
>
> **Our analyses on the unique aspects of temporal understanding with CD have not been covered by existing visual CD works.** It is the unique observations and analyses that led us to propose TDU, which consistently outperforms existing visual CD methods on temporal reasoning. Therefore, our work is not just a simple repurposing CD for video, but a targeted study of the unique challenges in temporal understanding and how to correctly and effectively address them with TDU.
>
>
>
> ## Weakness 2, Weakness 3
>
> Thanks for your advice! We added results of: InternVL3 (8B, 38B), QwenVL2.5 (7B, 72B), and LLaVA-Video-72B-Qwen2, with the same setting as LLaVA-Video-Qwen2 for evaluation. Tab. 1 shows our method also improves recent MLLMs such as InternVL3 and Qwen2.5-VL, and can scale to larger models (38B, 72B) on different benchmarks.
>
> | Model                 |      |            |      | Video-MME | MLVU     | LVBench  | MVBench  |
> | --------------------- | ---- | ---------- | ---- | --------- | -------- | -------- | -------- |
> | LLaVA-Video-7B-Qwen2  |      | Vanilla    |      | 59.8      | 67.4     | 58.2     | 58.5     |
> | LLaVA-Video-7B-Qwen2  |      | + VCD      |      | 60.0      | 67.3     | 58.3     | 58.7     |
> | LLaVA-Video-7B-Qwen2  |      | **+ Ours** |      | **61.6**  | **68.5** | **59.1** | **59.7** |
> | LLaVA-Video-72B-Qwen2 |      | Vanilla    |      | 67.5      | 71.2     | 61.9     | 64.1     |
> | LLaVA-Video-72B-Qwen2 |      | + VCD      |      | 67.6      | 72.2     | 61.9     | 64.3     |
> | LLaVA-Video-72B-Qwen2 |      | **+ Ours** |      | **68.8**  | **73.1** | **62.7** | **65.2** |
> | Qwen2.5-VL-7B         |      | Vanilla    |      | 65.1      | 70.2     | 45.3     | 69.6     |
> | Qwen2.5-VL-7B         |      | + VCD      |      | 65.3      | 70.3     | 45.3     | 69.7     |
> | Qwen2.5-VL-7B         |      | **+ Ours** |      | **66.7**  | **71.2** | **46.1** | **70.6** |
> | Qwen2.5-VL-72B        |      | Vanilla    |      | 73.3      | 74.6     | 47.3     | 70.4     |
> | Qwen2.5-VL-72B        |      | + VCD      |      | 73.3      | 74.6     | 47.2     | 70.6     |
> | Qwen2.5-VL-72B        |      | **+ Ours** |      | **74.5**  | **75.5** | **48.0** | **71.3** |
> | InternVL3-8B          |      | Vanilla    |      | 66.3      | 71.4     | 58.8     | 75.4     |
> | InternVL3-8B          |      | + VCD      |      | 66.4      | 71.4     | 58.8     | 75.5     |
> | InternVL3-8B          |      | **+ Ours** |      | **68.1**  | **72.5** | **59.6** | **76.7** |
> | InternVL3-38B         |      | Vanilla    |      | 72.7      | 77.8     | 67.3     | 76.9     |
> | InternVL3-38B         |      | + VCD      |      | 72.7      | 77.7     | 67.2     | 77.0     |
> | InternVL3-38B         |      | **+ Ours** |      | **73.8**  | **78.7** | **68.1** | **78.1** |
>
> (Tab. 1. More evaluation with different benchmarks and Video LLMs)
>
>
>
> ## Weakness 4, Question 2
>
> Thanks for the insightful question. Due to the nature of Contrastive Decoding (which requires two forward passes), CD-based methods (e.g., VCD and ours) are inevitably slower than the vanilla model in generation speed.
>
> In practical scenarios, it's difficult to optimize performance, time efficiency, and memory efficiency all at once. Different applications prioritize different aspects. CD-based methods are less time-efficient, but they offer better performance without a significant increase in **memory usage**. This makes them well-suited for applications where content quality is critical, such as education and medical assistance, where accuracy really matters and errors come at a high cost.
>
> From Tab. 2, we observed that the time and memory efficiency of our TDU+CD is similar to VCD. CD based models (ours and VCD) are inevitably slower than the vanilla model due to CD'2 nature, but the additional memory usage is efficent, especially when model size grows.
>
> |                                      | LLaVA-Video-7B-Qwen2 | LLaVA-Video-7B-Qwen2 | LLaVA-Video-7B-Qwen2 | LLaVA-Video-72B-Qwen2 | LLaVA-Video-72B-Qwen2 | LLaVA-Video-72B-Qwen2 |
> | ------------------------------------ | -------------------- | -------------------- | -------------------- | --------------------- | --------------------- | --------------------- |
> | **Metric**                           | Vanilla              | + VCD                | + Ours               | Vanilla               | + VCD                 | + Ours                |
> | Avg Inference Latency per sample (s) | 1.026                | 1.670                | 1.675                | 5.315                 | 9.162                 | 9.165                 |
> | Avg Tokens per sec (tokens/s)        | 3.52                 | 2.107                | 2.101                | 0.679                 | 0.394                 | 0.394                 |
> | Avg GPU Memory Usage (MiB)           | 41418                | 43986                | 43988                | 193261                | 199590                | 199596                |
> | Total GPU Hours (h)                  | 1.144                | 1.862                | 1.867                | 6.034                 | 10.223                | 10.227                |
>
> (Tab. 2. Inference Statistics on A100 80GB on TempCompass)
>
> For CD, its time efficiency might be improved through techniques such as leveraging the KV cache in some way, token pruning, or early stopping in the first forward pass, etc. This presents an interesting future research direction. However, our focus is not on optimizing the speed of CD algorithms, but rather on designing CD-based methods that effectively enhance temporal reasoning. Our guiding principle is to ensure that **our approach does not show inference performance degradation compared to common CD based methods like VCD**, rather than aiming to match the efficiency of the vanilla model. It is a separate and challenging reseach area different from our goal of temporal resoning.
>
> > a 2–3 % boost may not justify the extra cost in real-time settings ... a larger models (14B) often yield ≥ 2 % improvements out-of-the-box compared to a 7B model.
>
> Thanks for the valuable opinion. Compared to 7B/8B models with TDU+CD, Tab. 2 shows that larger vanilla models (e.g., 38B and 72B) indeed exhibit significant better results. However, it is noted that when these larger models are combined with our TDU+CD approach, **their performance can be further improved**. Our goal is not to make smaller models outperform much larger ones, which is almost impossible in this task given the substantial initial performance gap, but rather to **enhance the capabilities of models at different scale**. Our method achieves this effectively.
>
> For example, LLaVA-Video-72B-Qwen2 and Qwen2.5-VL-72B are already the largest models in their respective series, and there are no larger variants available within those model families. Yet, even at this scale, applying TDU+CD still brings improvements. Furthermore, in many practical scenarios where GPU resources are limited, increasing model size is not a viable option for improving accuracy. In such cases, TDU+CD offers a practical and efficient alternative for boosting performance without notably scaling up the model size.
>
>
>
> ## Question 3
>
> Different distortion ratios do have an impact on model performance. With slight adjustments based on the task, the model can achieve best results. Intuitively, different videos or tasks have varying levels of information density, so using different distortion ratios allows better adaptation to specific scenarios. However, significant per-task tuning is not necessary to achieve improvements. As long as the ratio is not set to an extreme (i.e., very close to 0 or 1), simply choosing a moderate value around 0.5 already leads to consistent gains across different tasks. We show such setting (Ratio=0.5) on LLaVA-Video-Qwen2-7B on different tasks in Tab. 3.
>
> For other parameters such as window size, block size, alpha, and beta, we use the **same** values across all tasks without any per-task tuning.
>
> |                | Vanilla Model | Best Setting | Ratio=0.5 |
> | -------------- | ------------- | ------------ | --------- |
> | EventHallusion | 63.5          | 72.1         | 71.3      |
> | TempCompass    | 65.0          | 67.5         | 66.7      |
> | MVBench        | 58.5          | 59.7         | 59.2      |
> | Video-MME      | 59.8          | 61.6         | 61.1      |
>
> (Tab. 3. Different ratio settings)
>
> We sincerely thank you again for your review. Your comments are very helpful. We hope our responses can address your concerns. If you have any further questions or unresolved issues, we are glad to have further discussions!

---

> ### Author Response · Authors · 2025-08-05
>
> We sincerely thank you again for your review! As the Reviewer-Author Discussion period is coming to a close, we would like to kindly ask whether our responses have addressed your concerns? We will be grateful for any feedback from you!

---

> > ### Comment · Reviewer_SAQe · 2025-08-05
> >
> > I appreciate the authors' clarification and the additional experimental results. However, I still find the proposed method has three major limitations: 1. Limited novelty: The contrastive distillation (CD) framework for visual understanding has been well-established in prior work. The authors’ contribution—replacing noise-based negatives with distorted video frames—is incremental and does not substantially advance the core idea. 2. Limited generalizability: The method requires task-specific hyperparameter tuning, which limits its applicability in real-world scenarios. Given this constraint, it is unlikely that inference-time model providers would adopt this method for serving general-purpose video understanding models. 3. Cost-effectiveness: The authors have not justified whether the performance gains from using CD outweigh the roughly 2× increase in test time compute. In particular, a direct comparison between a smaller model with CD (e.g., 7B) and a larger model without CD (e.g., 14B) is still necessary. The author can argue their method can push the boundary of the largest model. But there are also many other methods that increase the model performance at the cost of test time compute such as test time scaling. There is not such discussion in the paper.
> >
> > Overall, I find this paper has limited potential impact. I would like to keep my rating.

---

> > > ### Author Response · Authors · 2025-08-05
> > >
> > > We thank you for your feedback.
> > >
> > > ### 1. Limited Novelty
> > > > Limited novelty...The authors’ contribution—replacing noise-based negatives with distorted video frames—is incremental and does not substantially advance the core idea.
> > >
> > > We kindly introduced in responses to *Weakness 1* that, the distortion design is not merely an incremental improvement, instead, **the core of CD based methods, is how to find and create effective negatives targets (i.e., how to distort)**.
> > >
> > > Our contribution goes beyond prior visual CD frameworks in the following ways:
> > >
> > > 1. We introduce a novel analysis of temporal understanding, identifying key factors, language and image priors, that impact reasoning (Sec. 2).
> > > 2. We propose the new temporal distortion methodology based on essential conditions to mitigate their biases effectively (Sec. 3.2), which together offer new insights into CD in video temporal understanding.
> > >
> > > The results also support this point: the prior visual CD shows **very little to no improvement** on video tasks, whereas our method achieves a significant performance gain.
> > >
> > > ### 2. Limited Generalizability
> > > > Limited generalizability: The method requires task-specific hyperparameter tuning, which limits its applicability in real-world scenarios.
> > >
> > > As we introduced in responses to *Question 3* (also in Tab. 3), task-specific hyperparameter tuning is **not** necessary.
> > > 1. For distortion ratio parameters, we show in Tab. 3 that, as long as the ratio is not set to an extreme (i.e., very close to 0 or 1), simply choosing a moderate value around 0.5 for all already leads to consistent gains across different tasks, including temporal-centric task **TemporalCompass**, event-centric task **EventHallusion**, as well as general video understanding tasks **Video-MME** and **MVBench**.
> > >
> > > 2. For other parameters such as window size, block size, alpha, and beta, we use the **same values across all tasks without any per-task tuning.**
> > >
> > > Therefore, our method can generalize to different real-world scenarios for general-purposes. We kindly ask if it could address your concern regarding generalizability?
> > >
> > > ### 3. Cost Effectiveness
> > > >  The authors have not justified whether the performance gains from using CD outweigh the roughly 2× increase in test time. In particular, a direct comparison between a smaller model with CD (e.g., 7B) and a larger model without CD (e.g., 14B) is still necessary.
> > >
> > > This comparison is in fact not fair, as 14B or larger models also require two times or even more memory overhead. In contrast, our method introduces almost **no additional memory cost**. Still, on three representative general video understanding benchmarks, our method slightly outperforms InternVL3-14B on MVBench, and on Video-MME and MLVU, our method reduces the performance gap to InternVL3-14B by nearly half.
> > >
> > > | Model             | Method     | Video-MME | MLVU  | MVBench |
> > > |------------------|------------|-----------|-------|---------|
> > > | InternVL3-8B     | Vanilla    | 66.3      | 71.4  | 75.4    |
> > > | InternVL3-8B     | + VCD      | 66.4      | 71.4  | 75.5    |
> > > | InternVL3-8B     | + Ours     | 68.1      | 72.5  | **76.7**    |
> > > | InternVL3-14B    | Vanilla    | **70.4**      | **73.3**  | 76.6    |
> > >
> > > > The author can argue their method can push the boundary of the largest model. But there are also many other methods that increase the model performance at the cost of test time compute such as test time scaling. There is not such discussion in the paper
> > >
> > > Typical test-time scaling strategies, such as using Chain-of-Thought (CoT) prompting, Retrieval-Augmented Generation (RAG), or Mixture-of-Experts (MoE), etc. , are orthogonal to our approach. Our decoding-based method can be applied on test time scaling methods and serves as a further enhancement to these techniques, rather than a direct counterpart or competitor.

---

> > > ### Author Response · Authors · 2025-08-08
> > > **Follow-up on Effectiveness: We doubled our speed with very simple torch.cuda.Stream()**
> > >
> > > We highly value your feedback and have been working hard to address it.
> > >
> > > Regarding your concern about the inference time of Contrastive Decoding-based methods (including ours), we have simply optimized the code to **reduce the time by nearly half with no impact on the results.** **It is now as efficient as the vanllia Video LLM**
> > >
> > > The improvement is simple: we parallelized the two forward passes during next-token generation.
> > > Previously, we implemented it in the `def sample()` function as follows:
> > >
> > > ```python
> > > outputs = self.forward(...)       # inference with raw video
> > > outputs_cd = self.forward(...)    # inference with distorted video
> > >
> > > # calculate final logits with outputs and outputs_cd:
> > > ...
> > > ```
> > >
> > > Now, with just a few extra lines, we use `torch.cuda.Stream()` to run both forward passes in parallel:
> > >
> > > ```python
> > > stream_main = torch.cuda.Stream()
> > > stream_cd = torch.cuda.Stream()
> > > outputs_holder = {}
> > > outputs_cd_holder = {}
> > >
> > > # submit main inference
> > > with torch.cuda.stream(stream_main):
> > >     outputs_holder['main'] = self.forward(...)
> > >
> > > # submit contrastive inference
> > > with torch.cuda.stream(stream_cd):
> > >     outputs_cd_holder['cd'] = self.forward(...)
> > >
> > > torch.cuda.synchronize()
> > >
> > > outputs = outputs_holder['main']
> > > outputs_cd = outputs_cd_holder['cd']
> > >
> > > # calculate final logits with outputs and outputs_cd:
> > > ...
> > > ```
> > >
> > > Here is the comparison of parallel version with the original version:
> > >
> > > | Metric                              | LLaVA-Video-7B-Qwen2 Vanilla | LLaVA-Video-7B-Qwen2 + VCD | LLaVA-Video-7B-Qwen2 + Ours | LLaVA-Video-7B-Qwen2 + Ours (**Parallel**) |
> > > |-------------------------------------|-------------------------------|-----------------------------|------------------------------|------------------------------------------|
> > > | Avg Inference Latency per sample (s)| 1.026                         | 1.670                       | 1.675                        | **1.029**                                    |
> > > | Avg Tokens per sec (tokens/s)       | 3.52                          | 2.107                       | 2.101                        | **3.50**                                     |
> > > | Avg GPU Memory Usage (MiB)          | 41418                         | 43986                       | 43988                        | **41422**                                    |
> > > | Total GPU Hours (h)                 | 1.144                         | 1.862                       | 1.867                        | **1.148**                              |
> > >
> > >
> > > Our method is as fast as the vanllia Video LLM with the same **average** GPU memory usage. The only cost is about a 20% increase in *peak* GPU memory usage.
> > >
> > > We thank you again for your time and efforts! Would you consider our method efficient as well? We hope our responses can help to alleviate your remaining concerns.

---

> > > ### Author Response · Authors · 2025-08-08
> > >
> > > As the discussion phase is coming to a close, we would like to discuss whether our additional clarifications and results can help to address your remaining concerns!
> > >
> > > We would be happy to discuss further if you have any additional questions or feedback. Thank you once again for your detailed and thoughtful review and suggestions!

---

### Official Review · Reviewer_H8xN · 2025-06-22

**Clarity:** 3
**Significance:** 3
**Originality:** 3
**Rating:** 4
**Confidence:** 3

**Summary:**

This paper proposes a training-free method to improve temporal reasoning in Video Large Language Models (VLLMs). By intentionally distorting the temporal order of key frames during decoding and contrastively avoiding the resulting incorrect outputs, the model is guided toward more temporally coherent predictions.

**Questions:**

See Weaknesses

**Ethical Concerns:**

["NO or VERY MINOR ethics concerns only"]

**Final Justification:**

After reading the authors' responses, most of my concerns are addressed. I decided to maintain my positive score.

**Limitations:**

See Weaknesses

**Quality:**

3

**Strengths And Weaknesses:**

Strengths:
1. The method is architecture-agnostic and can be applied to multiple video prediction models, showing improvements across diverse datasets and backbones.
2. The proposed method yields consistent improvements across both temporal-specific and general video understanding benchmarks.

Weaknesses:
1. **Inference Overhead Not Addressed**:
Although the method is training-free, it introduces additional computational cost during inference. A quantitative comparison of inference time with other contrastive decoding methods would better justify its efficiency.

2. **Missing Failure Case Analysis**:
The paper does not present any failure cases or qualitative error analysis. Including such examples would help clarify the method’s limitations and guide future improvements.

3. **Limited Baseline Comparisons**:
The experimental evaluation lacks comparisons with more recent MLLM baselines, such as InternVL3. Including stronger and more up-to-date models would provide a more comprehensive assessment of the method’s effectiveness.

---

> ### Author Rebuttal · Authors · 2025-07-30
>
> Thank you for your review and suggestions!
>
> ## Weakness 1
>
> Thanks for the meaningful advice! Due to the nature of Contrastive Decoding (which requires two forward passes), CD-based methods (e.g., VCD and ours) are inevitably slower than the vanilla model in generation speed.
>
> In practical scenarios, it's difficult to optimize performance, time efficiency, and memory efficiency all at once. Different applications prioritize different aspects. CD-based methods are less time-efficient, but they offer better performance without a significant increase in memory usage. This makes them well-suited for applications where content quality is critical, such as education and medical assistance, where accuracy really matters and errors come at a high cost.
>
> From Tab. 1, we observed that the time and memory efficiency of our TDU+CD is similar to VCD. CD based models (ours and VCD) are inevitably slower than the vanilla model due to CD's nature, but the additional memory usage is efficent, especially when model size grows.
>
> |                                      | LLaVA-Video-7B-Qwen2 | LLaVA-Video-7B-Qwen2 | LLaVA-Video-7B-Qwen2 | LLaVA-Video-72B-Qwen2 | LLaVA-Video-72B-Qwen2 | LLaVA-Video-72B-Qwen2 |
> | ------------------------------------ | -------------------- | -------------------- | -------------------- | --------------------- | --------------------- | --------------------- |
> | **Metric**                           | Vanilla              | + VCD                | + Ours               | Vanilla               | + VCD                 | + Ours                |
> | Avg Inference Latency per sample (s) | 1.026                | 1.670                | 1.675                | 5.315                 | 9.162                 | 9.165                 |
> | Avg Tokens per sec (tokens/s)        | 3.52                 | 2.107                | 2.101                | 0.679                 | 0.394                 | 0.394                 |
> | Avg GPU Memory Usage (MiB)           | 41418                | 43986                | 43988                | 193261                | 199590                | 199596                |
> | Total GPU Hours (h)                  | 1.144                | 1.862                | 1.867                | 6.034                 | 10.223                | 10.227                |
>
> (Tab. 1. Inference Statistics on A100 80GB on TempCompass)
>
> For CD, its time efficiency might be improved through techniques such as leveraging the KV cache in some way, token pruning, or early stopping in the first forward pass, etc. This presents an interesting future research direction. However, our focus is not on optimizing the speed of CD algorithms, but rather on designing CD-based methods that effectively enhance temporal reasoning. Our guiding principle is to ensure that **our approach does not show inference performance degradation compared to common CD based methods like VCD**, rather than aiming to match the efficiency of the vanilla model. It is a separate and challenging reseach area different from our goal of temporal resoning.
>
>
>
> ## Weakness 2
>
> Thanks for the suggestions! In Appendix D (Limitations), we have discussed the possible weaknesses and failures of our method, as well as possible future directions.
>
> Our method tends to show relatively degraded performance when the input video is highly complex or extremely long. A key underlying reason is that attention maps do not always accurately capture the true importance of each frame and visual token, especially when the input videos are intricate as mentioned above. In such cases, their reliability further deteriorates, and relying on them often leads to only coarse-grained results.
>
> We will include more examples in the final version to better illustrate the analysis!
>
>
>
> ## Weakness 3
>
> Thanks for your advice! We added results of: InternVL3 (8B, 38B), QwenVL2.5 (7B, 72B), and LLaVA-Video-72B-Qwen2 for evaluation. Tab. 2 shows our method also improves recent MLLMs such as InternVL3 and Qwen2.5-VL, and can scale to larger models (38B, 72B) on different benchmarks.
>
> | Model                 |      |            |      | Video-MME | MLVU     | LVBench  | MVBench  |
> | --------------------- | ---- | ---------- | ---- | --------- | -------- | -------- | -------- |
> | LLaVA-Video-7B-Qwen2  |      | Vanilla    |      | 59.8      | 67.4     | 58.2     | 58.5     |
> | LLaVA-Video-7B-Qwen2  |      | + VCD      |      | 60.0      | 67.3     | 58.3     | 58.7     |
> | LLaVA-Video-7B-Qwen2  |      | **+ Ours** |      | **61.6**  | **68.5** | **59.1** | **59.7** |
> | LLaVA-Video-72B-Qwen2 |      | Vanilla    |      | 67.5      | 71.2     | 61.9     | 64.1     |
> | LLaVA-Video-72B-Qwen2 |      | + VCD      |      | 67.6      | 72.2     | 61.9     | 64.3     |
> | LLaVA-Video-72B-Qwen2 |      | **+ Ours** |      | **68.8**  | **73.1** | **62.7** | **65.2** |
> | Qwen2.5-VL-7B         |      | Vanilla    |      | 65.1      | 70.2     | 45.3     | 69.6     |
> | Qwen2.5-VL-7B         |      | + VCD      |      | 65.3      | 70.3     | 45.3     | 69.7     |
> | Qwen2.5-VL-7B         |      | **+ Ours** |      | **66.7**  | **71.2** | **46.1** | **70.6** |
> | Qwen2.5-VL-72B        |      | Vanilla    |      | 73.3      | 74.6     | 47.3     | 70.4     |
> | Qwen2.5-VL-72B        |      | + VCD      |      | 73.3      | 74.6     | 47.2     | 70.6     |
> | Qwen2.5-VL-72B        |      | **+ Ours** |      | **74.5**  | **75.5** | **48.0** | **71.3** |
> | InternVL3-8B          |      | Vanilla    |      | 66.3      | 71.4     | 58.8     | 75.4     |
> | InternVL3-8B          |      | + VCD      |      | 66.4      | 71.4     | 58.8     | 75.5     |
> | InternVL3-8B          |      | **+ Ours** |      | **68.1**  | **72.5** | **59.6** | **76.7** |
> | InternVL3-38B         |      | Vanilla    |      | 72.7      | 77.8     | 67.3     | 76.9     |
> | InternVL3-38B         |      | + VCD      |      | 72.7      | 77.7     | 67.2     | 77.0     |
> | InternVL3-38B         |      | **+ Ours** |      | **73.8**  | **78.7** | **68.1** | **78.1** |
>
> (Tab. 2. More evaluation with different benchmarks and models)
>
> We sincerely thank you again for your review. Your comments are very helpful. We hope our responses can address your concerns. If you have any further questions or unresolved issues, we are glad to have further discussions!

---

> > ### Comment · Reviewer_H8xN · 2025-08-06
> > **Official Comment by Reviewer H8xN**
> >
> > Thank you for the thorough rebuttal and the additional experiments. I have carefully reconsidered my initial score in light of the author's responses and the discussions. While the paper is technically sound and clearly written, I find the overall contribution incremental relative to prior contrastive decoding (CD) frameworks, and the inference-time cost remains insufficiently justified by the reported gains from other reviewers' comments. Given the above, I may decline my score to 3.

---

> > > ### Author Response · Authors · 2025-08-07
> > >
> > > We thank you for your feedback!
> > >
> > > >  I find the overall contribution incremental relative to prior contrastive decoding (CD) frameworks
> > >
> > > Our method is not merely a straightforward extension to CD. In fact, **the core of CD based methods, is how to find and create effective negatives targets (i.e., how to distort)**.
> > >
> > > Our contribution goes beyond prior visual CD frameworks in the following ways:
> > >
> > > 1. We introduce a novel analysis of temporal understanding, identifying key factors, language and image priors, that impact reasoning (Sec. 2).
> > > 2. We propose the new temporal distortion methodology based on essential conditions to mitigate their biases effectively (Sec. 3.2), which together offer new insights into CD in video temporal understanding.
> > >
> > > The results also support this point: the prior visual CD shows **very little or no improvement** on video tasks, whereas our method achieves a notable performance gain.
> > >
> > >
> > > > the inference-time cost remains insufficiently
> > >
> > > In practical scenarios, it's difficult to optimize performance, time efficiency, and memory efficiency all at once. Different applications prioritize different aspects.
> > >
> > > CD-based methods are less time-efficient, but they offer better performance **without a significant increase in memory usage**. This makes them well-suited for applications where content quality is critical, such as education and medical assistance, where accuracy really matters and errors come at a high cost.  From Tab. 2, we observed that the time and memory efficiency of our TDU+CD is similar to VCD. CD based models (ours and VCD) are inevitably slower than the vanilla model due to CD's nature, but **the additional memory usage is efficent, especially when model size grows.**
> > >
> > > ### Tab. 2. Inference Statistics on A100 80GB on TempCompass
> > >
> > > | Model                         | Vanilla | + VCD | + Ours | Vanilla | + VCD | + Ours |
> > > |------------------------------|---------|-------|--------|---------|-------|--------|
> > > |                              | LLaVA-Video-7B-Qwen2 | LLaVA-Video-7B-Qwen2 | LLaVA-Video-7B-Qwen2 | LLaVA-Video-72B-Qwen2 | LLaVA-Video-72B-Qwen2 | LLaVA-Video-72B-Qwen2 |
> > > | Avg Inference Latency per sample (s) | 1.026 | 1.670 | 1.675 | 5.315 | 9.162 | 9.165 |
> > > | Avg Tokens per sec (tokens/s) | 3.52 | 2.107 | 2.101 | 0.679 | 0.394 | 0.394 |
> > > | Avg GPU Memory Usage (MiB) | 41418 | 43986 | 43988 | 193261 | 199590 | 199596 |
> > > | Total GPU Hours (h) | 1.144 | 1.862 | 1.867 | 6.034 | 10.223 | 10.227 |
> > >
> > > Thank you again for your thoughtful review. We hope our further clarifications can help to alleviate your remaining concerns!

---

> > > ### Author Response · Authors · 2025-08-08
> > > **Follow-up on Effectiveness: We doubled our speed with very simple torch.cuda.Stream()**
> > >
> > > We highly value your feedback and have been working hard to address it.
> > >
> > > Regarding your concern about the inference time of Contrastive Decoding-based methods (including ours), we have simply optimized the code to **reduce the time by nearly half with no impact on the results.** **It is now as efficient as the vanllia Video LLM**
> > >
> > > The improvement is simple: we parallelized the two forward passes during next-token generation.
> > > Previously, we implemented it in the `def sample()` function as follows:
> > >
> > > ```python
> > > outputs = self.forward(...)       # inference with raw video
> > > outputs_cd = self.forward(...)    # inference with distorted video
> > >
> > > # calculate final logits with outputs and outputs_cd:
> > > ...
> > > ```
> > >
> > > Now, with just a few extra lines, we use `torch.cuda.Stream()` to run both forward passes in parallel:
> > >
> > > ```python
> > > stream_main = torch.cuda.Stream()
> > > stream_cd = torch.cuda.Stream()
> > > outputs_holder = {}
> > > outputs_cd_holder = {}
> > >
> > > # submit main inference
> > > with torch.cuda.stream(stream_main):
> > >     outputs_holder['main'] = self.forward(...)
> > >
> > > # submit contrastive inference
> > > with torch.cuda.stream(stream_cd):
> > >     outputs_cd_holder['cd'] = self.forward(...)
> > >
> > > torch.cuda.synchronize()
> > >
> > > outputs = outputs_holder['main']
> > > outputs_cd = outputs_cd_holder['cd']
> > >
> > > # calculate final logits with outputs and outputs_cd:
> > > ...
> > > ```
> > >
> > > Here is the comparison of parallel version with the original version:
> > >
> > > | Metric                              | LLaVA-Video-7B-Qwen2 Vanilla | LLaVA-Video-7B-Qwen2 + VCD | LLaVA-Video-7B-Qwen2 + Ours | LLaVA-Video-7B-Qwen2 + Ours (**Parallel**) |
> > > |-------------------------------------|-------------------------------|-----------------------------|------------------------------|------------------------------------------|
> > > | Avg Inference Latency per sample (s)| 1.026                         | 1.670                       | 1.675                        | **1.029**                                    |
> > > | Avg Tokens per sec (tokens/s)       | 3.52                          | 2.107                       | 2.101                        | **3.50**                                     |
> > > | Avg GPU Memory Usage (MiB)          | 41418                         | 43986                       | 43988                        | **41422**                                    |
> > > | Total GPU Hours (h)                 | 1.144                         | 1.862                       | 1.867                        | **1.148**                              |
> > >
> > >
> > > Our method is as fast as the vanllia Video LLM with the same **average** GPU memory usage. The only cost is about a 20% increase in *peak* GPU memory usage.
> > >
> > > We thank you again for your time and efforts! We hope our responses can help to alleviate your remaining concerns.

---

> > > > ### Comment · Reviewer_H8xN · 2025-08-09
> > > > **Official Comment by Reviewer H8xN**
> > > >
> > > > Thanks for the author's reply. I will reconsider the score.

---

> ### Author Response · Authors · 2025-08-05
>
> We sincerely thank you again for your review! As the Reviewer-Author Discussion period is coming to a close, we would like to kindly ask whether our responses have addressed your concerns? We will be grateful for any feedback from you!

---

### Official Review · Reviewer_mNF7 · 2025-07-07

**Clarity:** 3
**Significance:** 2
**Originality:** 3
**Rating:** 4
**Confidence:** 3

**Summary:**

This paper proposes a decoding method to improve temporal reasoning in Video LLMs by leveraging contrastive decoding with video temporal distortion. The core idea is to deliberately distort temporal consistency in key frames during decoding, which induces temporally insensitive wrong responses (e.g., based on text/image priors) from the model that can then be contrastively avoided during generation, guiding the model toward temporally grounded outputs. The method is applied to existing Video LLMs (e.g., LLaVA-Video-7B-Qwen2, Video-LLaVA), demonstrating consistent improvements on temporally-focused benchmarks (TempCompass, EventHallusion) and general video understanding benchmarks (Video-MME, MLVU), outperforming other contrastive decoding methods.

**Questions:**

Please check the questions in the weaknesses above.

**Ethical Concerns:**

["NO or VERY MINOR ethics concerns only"]

**Final Justification:**

After reading the authors' responses, most of my concerns are addressed. I decided to increase my score.

**Limitations:**

Yes

**Quality:**

2

**Strengths And Weaknesses:**

Strengths:

1. The paper introduces a simple and effective approach for improving temporal reasoning in Video LLMs without retraining, using contrastive decoding with targeted temporal distortion.

2. The proposed approach consistently improves performance on multiple temporal reasoning and general video understanding benchmarks, outperforming other contrastive decoding baselines.

Weaknesses:

1. While the identified “language prior” and “image prior” sound intuitively reasonable, the paper lacks empirical evidence to quantify how much these biases actually affect the model’s temporal reasoning performance. Providing concrete measurements would strengthen the motivation for the proposed approach.

2. Building on the previous point, while the paper proposes targeted temporal distortion methods motivated by these prior biases, the methods appear very heuristic and may heavily depend on the specific data distributions of the evaluated datasets. This raises concerns about the scalability and generalizability of the approach across different domains and video types.

3. Therefore, I would like to see the model’s performance on additional commonly used video understanding benchmarks, such as LVBench, EgoSchema, and MVBench, to better assess the generalization capability of the proposed method.

---

> ### Author Rebuttal · Authors · 2025-07-31
>
> Thank you for your review and suggestions!
>
> ## Weakness 1
>
> Thanks for the valuable advice! In fact, we provide a simple quantitative study in Sec. 2 (L91) to illustrate the impact of language priors. In this experiment, we analyze a randomly sampled subset of questions that Video-LLaVA and LLaVA-Video-7B-Qwen2 answered incorrectly on TempCompass, and measure how many of these were also answered incorrectly in a blind evaluation (where only the text questions are given without the video). This experiment highlights the correlation between language priors and temporal reasoning errors. We copied the results in Tab. 1.
>
> Results show that, among the incorrect predictions made by LLaVA-Video-7B-Qwen2 and Video-LLaVA, 46.7% and 38.9% respectively matched the answers produced by blind LLM,  significantly higher than random chance. It shows the correlation between language prior and temporal reasoning mistakes. In these cases, Video LLMs failed to ground their responses in videos and instead relied on language priors, leading to incorrect answers.
>
> |             | Video-LLaVA | LLaVA-Video-7B-Qwen2 |
> | ----------- | ----------- | -------------------- |
> | TempCompass | 46.7        | 38.9                 |
>
> (Tab. 1. Study on language prior)
>
> For image priors, we conducted a similar experiment. The difference is that we analyzed how many of the questions that Video-LLaVA and LLaVA-Video-7B-Qwen2 got wrong on TempCompass were also answered incorrectly in an image-only evaluation (where the model is given only the text questions and a randomly selected frame from the video, without access to the full video). This experiment aims to reveal the correlation between image priors and temporal reasoning errors.
>
> Similar to the results of the language prior experiment, the matching rates of 48.3% and 40.4% are significantly higher than random chance, indicating a strong correlation between image priors and temporal mistakes.
>
> |             | Video-LLaVA | LLaVA-Video-7B-Qwen2 |
> | ----------- | ----------- | -------------------- |
> | TempCompass | 48.3        | 40.4                 |
>
> (Tab. 2. Study on image prior)
>
> Together, these two experiments demonstrate a clear connection between language and image priors and temporal reasoning errors. This is an interesting research direction, and constructing a new dedicated benchmark may be required to perform a comprehensive and systematic analysis on it. However, our focus is not entirely on this aspect. Instead, we also aim to motivate from these observations and analyses to propose our methodology that can improve video temporal understanding.
>
>
>
> ## Weakness 2
>
> Thanks for the insightful question.
>
> > ... the methods appear very heuristic and may heavily depend on the specific data distributions of the evaluated datasets ...  This raises concerns about the scalability and generalizability of the approach across different domains and video types.
>
> In fact, generalizability and scalability are strengths of our approach. In essence, Contrastive Decoding (CD) works by generating bad answers and guiding the model to steer away from them to produce better results. In our case, we create bad answers by disrupting temporal information—specifically, by weakening temporal cues and emphasizing textual and static visual information in videos. This inherently ensures generalizability and scalability, as temporal information is one of the most fundamental and pervasive elements in video content. By disturbing it, we can consistently impair the model’s understanding across different types of videos and produce bad answers.
>
> Similarly, we added a study on the correlation between priors and reasoning mistakes on the MCQ subset of PerceptionTest that require the understanding of memory, abstract patterns, physics, and semantics etc. On PerceptionTest, we also calculated the matching rate (both language and image) that is proposed *weakness 1*. We can still see a clear correlation between language and image priors and mistakes.
>
> | PerceptionTest                 | Video-LLaVA | LLaVA-Video-7B-Qwen2 |
> | ------------------------------ | ----------- | -------------------- |
> | Matching Rate (language prior) | 43.1        | 37.5                 |
> | Matching Rate (image prior)    | 42.6        | 36.4                 |
>
> (Tab. 3. Matching Rate Results on PerceptionTest)
>
> |                            | LLaVA-Video-7B-Qwen2 | LLaVA-Video-7B-Qwen2 + VCD | LLaVA-Video-7B-Qwen2 + Ours |
> | -------------------------- | -------------------- | -------------------------- | --------------------------- |
> | PerceptionTest MCQ (Valid) | 60.18                | 60.57                      | **62.33**                   |
>
> (Tab. 4. Evaluation on PerceptionTest)
>
> Following your suggestion, we also performed evaluation on more video understanding benchmarks.
>
> ## Weakness 3
>
> Tab. 1, 2 and 3 quantitatively demonstrate the correlation between priors and temporal reasoning errors. In addition, following your suggestion, we incorporated commonly used benchmarks such as **LVBench**, **EgoSchema**, and **MVBench**. We also evaluated our method using larger 72B models and the most recent state-of-the-art image/video LLMs, Qwen2.5-VL. Our method consistently maintains a performance advantage on Tab. 5.
>
> | Model                 |          | Video-MME | MLVU     | EgoSchema | LVBench  | MVBench  |
> | --------------------- | -------- | --------- | -------- | --------- | -------- | -------- |
> | LLaVA-Video-7B-Qwen2  | Vanilla  | 59.8      | 67.4     | 57.3      | 58.2     | 58.5     |
> |                       | VCD      | 60.0      | 67.3     | 57.3      | 58.3     | 58.7     |
> |                       | Ours     | **61.6**  | **68.5** | **58.6**  | **59.1** | **59.7** |
> | LLaVA-Video-72B-Qwen2 | Vanilla  | 67.5      | 71.2     | 65.6      | 61.9     | 64.1     |
> |                       | VCD      | 67.6      | 72.2     | 65.7      | 61.9     | 64.3     |
> |                       | Ours     | **68.8**  | **73.1** | **66.9**  | **62.7** | **65.2** |
> | Qwen2.5-VL-7B         | Vanilla  | 65.1      | 70.2     | 65.0      | 45.3     | 69.6     |
> |                       | VCD      | 65.3      | 70.3     | 65.2      | 45.3     | 69.7     |
> |                       | **Ours** | **66.7**  | **71.2** | **66.5**  | **46.1** | **70.6** |
> | Qwen2.5-VL-72B        | Vanilla  | 73.3      | 74.6     | 76.2      | 47.3     | 70.4     |
> |                       | VCD      | 73.3      | 74.6     | 76.3      | 47.2     | 70.6     |
> |                       | **Ours** | **74.5**  | **75.5** | **77.2**  | **48.0** | **71.3** |
>
> (Tab. 5. More evaluation with different benchmarks and models)
>
> We sincerely thank you again for your review. Your advice are very helpful to us. We hope our responses can address your concerns. If you have any further questions or unresolved issues, we look forward to have further discussions!

---

> > ### Comment · Reviewer_mNF7 · 2025-08-05
> >
> > Thank you for the clarification and the additional experiments. They partially address my concerns. However, I still have remaining concerns regarding the paper's novelty. Given that the method appears to be a relatively straightforward extension of contrastive decoding, and considering both the inference overhead (mentioned by other reviewers) and the marginal improvements, I would like to maintain my current score.

---

> > > ### Author Response · Authors · 2025-08-05
> > >
> > > We thank you for your feedback.
> > >
> > > >  Given that the method appears to be a relatively straightforward extension of contrastive decoding
> > >
> > > Our method is not merely a straightforward extension to CD. In fact, **the core of CD based methods, is how to find and create effective negatives targets (i.e., how to distort)**.
> > >
> > > Our contribution goes beyond prior visual CD frameworks in the following ways:
> > >
> > > 1. We introduce a novel analysis of temporal understanding, identifying key factors, language and image priors, that impact reasoning (Sec. 2).
> > > 2. We propose the new temporal distortion methodology based on essential conditions to mitigate their biases effectively (Sec. 3.2), which together offer new insights into CD in video temporal understanding.
> > >
> > > The results also support this point: the prior visual CD shows **very little to no improvement** on video tasks, whereas our method achieves a significant performance gain.
> > >
> > >
> > >
> > > > Inference overhead (mentioned by other reviewers) and the marginal improvements
> > >
> > > In practical scenarios, it's difficult to optimize performance, time efficiency, and memory efficiency all at once. Different applications prioritize different aspects. CD-based methods are less time-efficient, but they offer better performance **without a significant increase in memory usage**. This makes them well-suited for applications where content quality is critical, such as education and medical assistance, where accuracy really matters and errors come at a high cost.  From Tab. 2, we observed that the time and memory efficiency of our TDU+CD is similar to VCD. CD based models (ours and VCD) are inevitably slower than the vanilla model due to CD'2 nature, but the additional memory usage is efficent, especially when model size grows.
> > >
> > > ### Tab. 2. Inference Statistics on A100 80GB on TempCompass
> > >
> > > | Model                         | Vanilla | + VCD | + Ours | Vanilla | + VCD | + Ours |
> > > |------------------------------|---------|-------|--------|---------|-------|--------|
> > > |                              | LLaVA-Video-7B-Qwen2 | LLaVA-Video-7B-Qwen2 | LLaVA-Video-7B-Qwen2 | LLaVA-Video-72B-Qwen2 | LLaVA-Video-72B-Qwen2 | LLaVA-Video-72B-Qwen2 |
> > > | Avg Inference Latency per sample (s) | 1.026 | 1.670 | 1.675 | 5.315 | 9.162 | 9.165 |
> > > | Avg Tokens per sec (tokens/s) | 3.52 | 2.107 | 2.101 | 0.679 | 0.394 | 0.394 |
> > > | Avg GPU Memory Usage (MiB) | 41418 | 43986 | 43988 | 193261 | 199590 | 199596 |
> > > | Total GPU Hours (h) | 1.144 | 1.862 | 1.867 | 6.034 | 10.223 | 10.227 |
> > >
> > >
> > > Our goal is not to make smaller models outperform much larger ones, which is almost impossible in this task given the substantial initial performance gap, but rather to enhance the capabilities of models at different scale. Our method achieves this effectively.
> > >
> > > For example, LLaVA-Video-72B-Qwen2 and Qwen2.5-VL-72B are already the largest models in their respective series, and there are no larger variants available within those model families. Yet, even at this scale, applying TDU+CD still brings improvements. Furthermore, in many practical scenarios where GPU resources are limited, increasing model size is not a viable option for improving accuracy. In such cases, TDU+CD offers a practical and efficient alternative for boosting performance without notably scaling up the model size.
> > >
> > > Thank you again for your thoughtful review. We hope our further clarifications help to alleviate your remaining concerns!

---

> > > ### Author Response · Authors · 2025-08-08
> > > **Follow-up on Inference Overhead (Effectiveness): We doubled our speed with very simple torch.cuda.Stream()**
> > >
> > > We highly value your feedback and have been working hard to address it.
> > >
> > > Regarding your concern about the inference time of Contrastive Decoding-based methods (including ours), we have simply optimized the code to **reduce the time by nearly half with no impact on the results.** **It is now as efficient as the vanllia Video LLM**
> > >
> > > The improvement is simple: we parallelized the two forward passes during next-token generation.
> > > Previously, we implemented it in the `def sample()` function as follows:
> > >
> > > ```python
> > > outputs = self.forward(...)       # inference with raw video
> > > outputs_cd = self.forward(...)    # inference with distorted video
> > >
> > > # calculate final logits with outputs and outputs_cd:
> > > ...
> > > ```
> > >
> > > Now, with just a few extra lines, we use `torch.cuda.Stream()` to run both forward passes in parallel:
> > >
> > > ```python
> > > stream_main = torch.cuda.Stream()
> > > stream_cd = torch.cuda.Stream()
> > > outputs_holder = {}
> > > outputs_cd_holder = {}
> > >
> > > # submit main inference
> > > with torch.cuda.stream(stream_main):
> > >     outputs_holder['main'] = self.forward(...)
> > >
> > > # submit contrastive inference
> > > with torch.cuda.stream(stream_cd):
> > >     outputs_cd_holder['cd'] = self.forward(...)
> > >
> > > torch.cuda.synchronize()
> > >
> > > outputs = outputs_holder['main']
> > > outputs_cd = outputs_cd_holder['cd']
> > >
> > > # calculate final logits with outputs and outputs_cd:
> > > ...
> > > ```
> > >
> > > Here is the comparison of parallel version with the original version:
> > >
> > > | Metric                              | LLaVA-Video-7B-Qwen2 Vanilla | LLaVA-Video-7B-Qwen2 + VCD | LLaVA-Video-7B-Qwen2 + Ours | LLaVA-Video-7B-Qwen2 + Ours (**Parallel**) |
> > > |-------------------------------------|-------------------------------|-----------------------------|------------------------------|------------------------------------------|
> > > | Avg Inference Latency per sample (s)| 1.026                         | 1.670                       | 1.675                        | **1.029**                                    |
> > > | Avg Tokens per sec (tokens/s)       | 3.52                          | 2.107                       | 2.101                        | **3.50**                                     |
> > > | Avg GPU Memory Usage (MiB)          | 41418                         | 43986                       | 43988                        | **41422**                                    |
> > > | Total GPU Hours (h)                 | 1.144                         | 1.862                       | 1.867                        | **1.148**                              |
> > >
> > >
> > > Our method is as fast as the vanllia Video LLM with the same **average** GPU memory usage. The only cost is about a 20% increase in *peak* GPU memory usage.
> > >
> > > We thank you again for your time and efforts! Would you consider our method efficient as well? We hope our responses can help to alleviate your remaining concerns.

---

> > > ### Author Response · Authors · 2025-08-08
> > >
> > > As the discussion phase is coming to a close, we would like to discuss whether our additional clarifications and results can help to address your remaining concerns!
> > >
> > > We would be happy to discuss further if you have any additional questions or feedback. Thank you once again for your detailed and thoughtful review and suggestions!

---

> ### Author Response · Authors · 2025-08-05
>
> We sincerely thank you again for your review! As the Reviewer-Author Discussion period is coming to a close, we would like to kindly ask whether our responses have addressed your concerns? We will be grateful for any feedback from you!

---

### Official Review · Reviewer_7wGw · 2025-07-23

**Clarity:** 3
**Significance:** 3
**Originality:** 3
**Rating:** 5
**Confidence:** 4

**Summary:**

This work attempts to improve the temporal understanding and reasoning abilities of modern Video Large Language Models (VLLMs). The authors first improve understanding of the ways in which Video LLMs fail by looking at the language and image priors which can dominate VLLM QA abilities due to the large amount of text and image data that is seen in pretraining. In order to fix these issues, the authors propose a modification tot he decoding phase of VLLMs (thus no additional training) to improve the temporal understanding capabilities of these models. Specifically, the authors propose a method to distort temporal consistency of key frames during decoding. These distortions produce time-insensitive incorrect model responses which can then be avoided in generating the final output via a contrastive decoding technique. The authors show this method improves temporal-specific and video understanding benchmark performance across many evaluations.

**Questions:**

See questions / concerns above in the Weaknesses section

**Ethical Concerns:**

["NO or VERY MINOR ethics concerns only"]

**Final Justification:**

I maintain my score of 5 (accept) as I believe this work is a valuable for improving our understanding of video understanding in VLMs and specifically demonstrates how to construct contrastive decoding examples for targeting temporal understanding in a relatively general way. I feel aligned with the authors that the other reviewer concerns are not sufficient grounds to reject the paper. Specifically, the other main concerns: increased inference time compute, and lack of novelty can be addressed by understanding this work in the context of the field of contrastive decoding and test-time compute for VLMs. It should not be held against the authors that contrastive decoding adds inference compute (this is a feature of all contrastive decoding methods and they do address this and show that their method strongly outperforms generic contrastive decoding). In addition, reviewers felt this work does not have strong novelty because it reuses visual contrastive decoding (which has already been introduced). However, like the authors say, the key aspect of contrastive decoding comes down to how you design the contrastive examples, and they clearly show that the prior generic visual contrastive decoding approaches to not give any benefits in temporal understanding tasks. On the other hand, their careful study of how to construct these examples gives insights into the features that VLMs use for video understanding and how we might bias them to focus on visual temporal features in QA tasks.

Overall, I think the work is well presented, has decently strong results and introduces a simple but effective method for improving the temporal understanding in VLMs without extra training. I feel this is enough to warrant acceptance.

**Limitations:**

yes

**Paper Formatting Concerns:**

No format concerns.

**Quality:**

3

**Strengths And Weaknesses:**

Strengths:
1. This work is relatively comprehensive in it’s understanding of how to distort videos in the best way to create effective negative examples during contrastive decoding. Unlike standard methods like just adding noise, random shuffling or dropping, they come up with a novel way to identify distortions that remove temporal information but still maintain salient spatial features of important frames. This is to my knowledge quite novel.
2. The writing is clear and the main ideas are well communicated
3. The authors evaluate on a reasonable set of benchmarks and use multiple video models to show the results generalize across at least 2.
4. The results compared to other contrastive decoding methods look to be significant.
5. The atuhors show that even on MLVU, on average the gains on temporal understanding outweigh slight losses in spatial understanding

Weaknesses:
1. There are quite a lot of hyperparameters here all shown in Figure 5. While the performance is somewhat stable across these, there are clear “best” choices and I don’t know with this many hyper params, how easy it would be for this method to generalize truly to all kinds of video LLMs or benchmark videos.
2. The authors could test on more benchmarks that are known to test failures in temporal perception like PerceptionTest and others. This woudl just strengthen the generality of the result.
3. From a practical perspective, the authors don’t address the extra inference compute needed to do this distortion generation and contrastive decoding at test-time. How does this practically impact the deployment of such a method for models as we scale the size of the LLMs.
4. There is no commentary I can see (unless I missed it) on statistical significance of the results. In particular how can we know that the gains from this method are statistically significant compared with the other contrastive decoding methods. For example, in the case where the distortion is just dropping random frames, could you run this multiple times to get a standard devation to compare?

---

> ### Author Rebuttal · Authors · 2025-07-30
>
> Thank you for your review and suggestions!
>
> ## Weakness 1
>
> Thanks for the valuable advice! Different distortion ratios do have an impact on model performance. With slight adjustments based on the task, the model can achieve best results. Intuitively, different videos or tasks have varying levels of information density, so using different distortion ratios allows better adaptation to specific scenarios. However, significant per-task tuning is not necessary to achieve improvements. As long as the ratio is not set to an extreme (i.e., very close to 0 or 1), simply choosing a moderate value around 0.5 already leads to consistent gains across different tasks. We show such setting (Ratio=0.5) on LLaVA-Video-Qwen2-7B on different tasks in Tab. 1.
>
> For other parameters such as window size, block size, alpha, and beta, we use the **same** values across **all tasks** and **models** without any per-task or model tuning.
>
> |                | Vanilla Model | Best Setting | Ratio=0.5 |
> | -------------- | ------------- | ------------ | --------- |
> | EventHallusion | 63.5          | 72.1         | 71.3      |
> | TempCompass    | 65.0          | 67.5         | 66.7      |
> | MVBench        | 58.5          | 59.7         | 59.2      |
> | Video-MME      | 59.8          | 61.6         | 61.1      |
>
> (Tab. 1. Ratio Setting)
>
>
>
> ## Weakness 2
>
> Thank you for the suggestion! To strengthen the generalizability of our results, we evaluated on a broader set of benchmarks (EgoSchema, LVBench, MVBench, and a multiple-choice subset of PerceptionTest-Valid), and included more recent models from the Qwen2.5-VL family and models at different sizes. As shown in Tab. 2, our method consistently outperforms baselines, demonstrating its generalizability and scalability.
>
> | Model                 |          | Video-MME | MLVU     | EgoSchema | LVBench  | MVBench  |
> | --------------------- | -------- | --------- | -------- | --------- | -------- | -------- |
> | LLaVA-Video-7B-Qwen2  | Vanilla  | 59.8      | 67.4     | 57.3      | 58.2     | 58.5     |
> |                       | VCD      | 60.0      | 67.3     | 57.3      | 58.3     | 58.7     |
> |                       | **Ours**     | **61.6**  | **68.5** | **58.6**  | **59.1** | **59.7** |
> | LLaVA-Video-72B-Qwen2 | Vanilla  | 67.5      | 71.2     | 65.6      | 61.9     | 64.1     |
> |                       | VCD      | 67.6      | 72.2     | 65.7      | 61.9     | 64.3     |
> |                       | **Ours**     | **68.8**  | **73.1** | **66.9**  | **62.7** | **65.2** |
> | Qwen2.5-VL-7B         | Vanilla  | 65.1      | 70.2     | 65.0      | 45.3     | 69.6     |
> |                       | VCD      | 65.3      | 70.3     | 65.2      | 45.3     | 69.7     |
> |                       | **Ours** | **66.7**  | **71.2** | **66.5**  | **46.1** | **70.6** |
> | Qwen2.5-VL-72B        | Vanilla  | 73.3      | 74.6     | 76.2      | 47.3     | 70.4     |
> |                       | VCD      | 73.3      | 74.6     | 76.3      | 47.2     | 70.6     |
> |                       | **Ours** | **74.5**  | **75.5** | **77.2**  | **48.0** | **71.3** |
>
> (Tab. 2. More evaluation with different benchmarks and models)
>
> Different from common recent Video LLM benchmarks such as EgoSchema, LVBench, and MVBench, PerceptionTest includes a broader set of tasks, including tracking and localization, which some Video LLMs cannot directly handle. To maintain consistency with prior benchmarks, we selected the multiple-choice question subset of PerceptionTest for evaluation. Results from Tab. 3 show our model also demonstrates clear advantages.
>
> |                             | LLaVA-Video-7B-Qwen2 | LLaVA-Video-7B-Qwen2 + VCD | LLaVA-Video-7B-Qwen2 + Ours |
> | --------------------------- | -------------------- | -------------------------- | --------------------------- |
> | PerceptionTest MCQ (Valid ) | 60.18                | 60.57                      | **62.33**                   |
>
> (Tab. 3. Evaluation on PerceptionTest)
>
>
>
> ## Weakness 3
>
> Thanks for the meaningful advice! Due to the nature of Contrastive Decoding (which requires two forward passes), CD-based methods (e.g., VCD and ours) are inevitably slower than the vanilla model in generation speed.
>
> In practical scenarios, it's difficult to optimize performance, time efficiency, and memory efficiency all at once. Different applications prioritize different aspects. CD-based methods are less time-efficient, but they offer better performance without a significant increase in **memory usage**. This makes them well-suited for applications where content quality is critical, such as education and medical assistance, where accuracy really matters and errors come at a high cost.
>
> From Tab. 4, we observed that the time and memory efficiency of our TDU+CD is similar to VCD. CD based models (ours and VCD) are inevitably slower than the vanilla model due to CD'2 nature, but the additional memory usage is efficent, especially when model size grows.
>
> |                                      | LLaVA-Video-7B-Qwen2 | LLaVA-Video-7B-Qwen2 | LLaVA-Video-7B-Qwen2 | LLaVA-Video-72B-Qwen2 | LLaVA-Video-72B-Qwen2 | LLaVA-Video-72B-Qwen2 |
> | ------------------------------------ | -------------------- | -------------------- | -------------------- | --------------------- | --------------------- | --------------------- |
> | **Metric**                           | Vanilla              | + VCD                | + Ours               | Vanilla               | + VCD                 | + Ours                |
> | Avg Inference Latency per sample (s) | 1.026                | 1.670                | 1.675                | 5.315                 | 9.162                 | 9.165                 |
> | Avg Tokens per sec (tokens/s)        | 3.52                 | 2.107                | 2.101                | 0.679                 | 0.394                 | 0.394                 |
> | Avg GPU Memory Usage (MiB)           | 41418                | 43986                | 43988                | 193261                | 199590                | 199596                |
> | Total GPU Hours (h)                  | 1.144                | 1.862                | 1.867                | 6.034                 | 10.223                | 10.227                |
>
> (Tab. 4. Inference Statistics on A100 80GB on TempCompass)
>
> For CD, its time efficiency might be improved through techniques such as leveraging the KV cache in some way, token pruning, or early stopping in the first forward pass, etc. This presents an interesting future research direction. However, our focus is not on optimizing the speed of CD algorithms, but rather on designing CD-based methods that effectively enhance temporal reasoning. Our guiding principle is to ensure that **our approach does not show inference performance degradation compared to common CD based methods like VCD**, rather than aiming to match the efficiency of the vanilla model. It is a separate and challenging reseach area different from our goal of temporal resoning.
>
>
>
> ## Weakness 4
>
> Thank you for advice. The results we reported in the paper are the average over three runs.
>
> Here we included results with standard deviations for reference:
>
> | Method                             | TempCompass | EventHallusion |
> | ---------------------------------- | ----------- | -------------- |
> | **Ours** (Full Method)             | 67.5 ± 0.21 | 72.1 ± 0.45    |
> | w/o **Attention Guidance**         | 65.7 ± 0.34 | 67.1 ± 0.71    |
> | w/o  **Frame Fusion**              | 66.1 ± 0.39 | 66.3 ± 0.73    |
> | w/o  **Token Distortion**          | 66.5 ± 0.41 | 68.3 ± 0.75    |
> | w/o  **Moving Content Disruption** | 66.9 ± 0.37 | 69.7 ± 0.69    |
> | VCD                                | 65.7 ± 0.49 | 66.5 ± 0.77    |
> | TCD                                | 65.7 ± 0.45 | 67.2 ± 0.79    |
>
> (Tab. 5. Results with standard deviations)
>
> We sincerely thank you again for your review. Your comments are very helpful. We hope our responses can address your concerns. If you have any further questions or unresolved issues, we are glad to have further discussions!

---

> > ### Comment · Reviewer_7wGw · 2025-08-05
> > **Response to rebuttal**
> >
> > I thank the authors for their response and additional experiments I encourage the authors to include the inference time latency comparison and discussions around that in the appendix of the paper (and refer to it in the main paper) as i believe that discussion is quite important for users to understand when and where to use this method.
> >
> > I appreciate the additional benchmarking and encourage the authors to include these results in the updated manuscript.
> >
> > I think this work is interesting and technically solid and thus maintain my score of 5 (Accept). I don't think the results warrant increasing any higher but I will maintain my score.

---

> > > ### Author Response · Authors · 2025-08-08
> > >
> > > We sincerely thank you again for your review, advice, and recognition of our work! We will include the additional results and discussion in the final version as you suggested!

---

### Note · Authors · 2025-08-11

We sincerely thank all SACs, ACs, and reviewers for your efforts in the review process!

We have addressed most of the reviewers’ concerns. Here we would like to bring our attention to the few remaining concerns after authors' **first** rebuttal and provide further clarifications.


### 1. Inference Effectiveness
Reviewer mNF7, H8xN, and SAQe expressed concerns on this point after the first rebuttal.

1. We would like to clarify that the longer inference time is **not** a drawback of our Video Distortion Method, but rather the nature of contrastive decoding based methods (which requires two forward passes).

2. Still, we further optimized the implementation and **it is now already as efficient as the vanilla Video LLM.**  Details are provided in our response to Reviewer SAQe under *Follow-up on Effectiveness*.

### 2. Novelty
We are not merely a straightforward extension of existing CD-based methods. In fact, the core of CD-based methods lies in *how to find and create effective negative targets to guide the decoding process (i.e., how to distort) .*

Our contributions go beyond prior visual CD frameworks in the following ways:

1. **Novel temporal understanding analysis.** We identify key factors, including language and image priors, that affect reasoning (Sec. 2).

   As recognized by:
   > This work is relatively comprehensive in its understanding of how to distort videos in the best way to create effective negative examples during contrastive decoding. (Reviewer 7wGw)

2. **New temporal distortion methodology.** We propose the new temporal distortion methodology based on essential conditions to mitigate their biases effectively (Sec. 3.2).

   As recognized by:
   > Unlike standard methods ... they come up with a novel way to identify distortions that remove temporal information but still maintain salient spatial features of important frames. *This is to my knowledge quite novel.* (Reviewer 7wGw)

Results further support this: prior visual CD methods show *very little to no* improvement on video tasks, whereas our method achieves a *notable* performance gain.

### 3. Generalizability

For the concern from Reviewer SAQe, we have showed that our method easily generalizes to various real-world, general-purpose scenarios in responses to Reviewer SAQe's *Question 3* with experiments.

We thank all SACs, ACs, and reviewers for your time and effort, and we hope the above clarifications would be helpful for you during your discussions!

---

### Decision · Program_Chairs · 2025-09-17

**Decision:**

Accept (poster)

**Comment:**

This paper investigates common failure cases in temporal reasoning with pre-trained video LLMs and proposes a simple solution combining targeted temporal distortion and contrastive decoding. Initial reviews were mixed, but after the rebuttal and discussion all reviewers recommended acceptance. Reviewers highlighted the technical innovation and strong empirical results, though two concerns were raised: (1) novelty over prior contrastive decoding methods, and (2) inference-time overhead. On the first, most reviewers agreed the work offers sufficient innovation despite building on contrastive decoding. On the second, the authors provided additional results addressing inference overhead during the discussion phase. After considering the paper, reviews, and discussion, the AC concurs with the reviewers’ post-rebuttal consensus and recommends acceptance. The authors are encouraged to incorporate the rebuttal clarifications into the camera-ready version.